# Overcoming Joint Intractability with Lossless Hierarchical Speculative Decoding

**Yuxuan Zhou**$^{\alpha,*}$, **Fei Huang**$^{\beta}$, **Heng Li**$^{\gamma}$, **Fengyi Wu**$^{\alpha}$, **Tianyu Wang**$^{\alpha}$,
**Jianwei Zhang**$^{\beta}$, **Junyang Lin**$^{\beta\dagger}$, **Zhi-Qi Cheng**$^{\alpha\dagger}$

$^{\alpha}$ University of Washington     $^{\gamma}$ Independent Researcher     $^{\beta}$ Qwen Team, Alibaba Inc.

`zhouyuxuanyx@gmail.com, junyang.ljy@alibaba-inc.com, zhiqics@uw.edu`

## Abstract

Verification is a key bottleneck in improving inference speed while maintaining distribution fidelity in Speculative Decoding. Recent work has shown that sequence-level verification leads to a higher number of accepted tokens compared to token-wise verification. However, existing solutions often rely on surrogate approximations or are constrained by partial information, struggling with joint intractability. In this work, we propose *Hierarchical Speculative Decoding (HSD)*, a provably lossless verification method that significantly boosts the expected number of accepted tokens and overcomes joint intractability by balancing excess and deficient probability mass across accessible branches. Our extensive large-scale experiments demonstrate that HSD yields consistent improvements in acceptance rates across diverse model families and benchmarks. Moreover, its strong explainability and generality make it readily integrable into a wide range of speculative decoding frameworks. Notably, integrating HSD into EAGLE-3 yields over a 12% performance gain, establishing state-of-the-art decoding efficiency without compromising distribution fidelity. Code is available at https://github.com/ZhouYuxuanYX/Hierarchical-Speculative-Decoding.

## 1 Introduction

Inference speed has become paramount for Large Language Models (LLMs) (Achiam et al., 2023; Touvron et al., 2023; Bai et al., 2023), which generate text auto-regressively. Recent advances in test-time scaling (OpenAI, 2024; Guo et al., 2025; Yu et al., 2025; Peng et al., 2025) have further underscored its importance. While techniques like pruning (Frankle and Carbin, 2018; Sun et al., 2023a) and quantization (Shen et al., 2020; Xiao et al., 2023) improve efficiency but sacrifice performance, Speculative Decoding (Leviathan et al., 2023) achieves speedups while preserving the target model's distribution, making it a particularly appealing alternative. It adopts a smaller model to make proposals and a larger model to select from them with a grounded verification strategy. Most approaches prioritize the drafting phase, but further gains face diminishing returns. Driven by the verification bottleneck, recent methods (Cai et al., 2024; Zhou et al., 2024; Narasimhan et al., 2024) trade off fidelity for speed, relying on task-specific tuning; their performance typically remains constrained to carefully curated scenarios.

Recent work (Sun et al., 2024; Qin et al., 2025) shows that jointly verifying draft tokens can improve the expected number of accepted tokens, but faces joint intractability: simply applying the resampling strategy used in tokenwise verification (Leviathan et al., 2023) would require full joint probabilities over all possible decoding paths to correctly recover the target distribution, which is computationally infeasible. To address this, (Qin et al., 2025) employs a lossy fixed acceptance threshold, while (Sun et al., 2024) proposes Blockwise Verification, which provably recovers the target distribution. However, Blockwise Verification still falls short of the ideal case, and both its underlying mechanism and compatibility with other methods remain unclear.

In this work, we propose Hierarchical Speculative Decoding (HSD), a provably lossless verification method built upon a novel hierarchical branch resampling strategy. In speculative decoding, resam-

---

$^{*}$Work done during internship at Qwen Team, Alibaba Inc.

$^{\dagger}$Corresponding author.

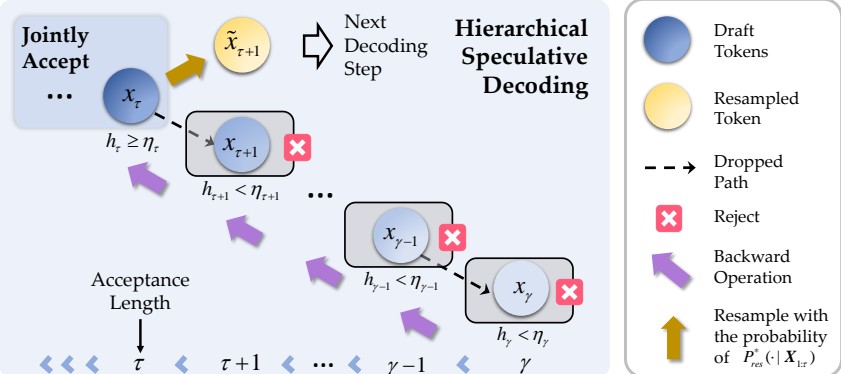

Figure 1: **Overview of HSD**. HSD accepts the draft $X_\tau$ by scanning backward from $\gamma$ to $\tau$, and then performs a single resampling at position $\tau + 1$ using the corresponding distribution from the resampling hierarchy.

pling recovers portions of the target distribution that exceed the draft probability. As illustrated in Figure 1, HSD organizes multiple resampling distributions hierarchically across successive levels, with each distribution recovering only the partial target within its branch and resampling occurring immediately after the last accepted token. This design ensures the full target distribution is recovered in expectation while maximizing the expected number of accepted tokens, pushing the limits of lossless verification and enabling more efficient decoding. Notably, Blockwise verification focuses on independent verification with unclear potential for integration, while our method is designed to easily combine with other approaches, such as the widely adopted multi-draft setups.

**In summary, our contributions are as follows**:

- We introduce *Hierarchical Speculative Decoding* (HSD), a lossless and explainable verification method that integrates seamlessly with existing speculative decoding frameworks while remaining largely orthogonal to them.

- HSD delivers a practical advance in inference scaling, achieving an average $6.7\%$ improvement in decoding speed across diverse benchmarks and model sizes while preserving distributional fidelity, with efficiency gains of up to $12.3\%$ on individual datasets.

- HSD further improves decoding speed across multi-draft settings. Notably, integrating HSD into EAGLE-3 yields over $12\%$ performance gain, establishing new state-of-the-art decoding efficiency without compromising distribution fidelity.

## 2 RELATED WORK

Follow-up research on speculative decoding (Leviathan et al., 2023) can be organized into two main phases: the drafting phase and the verification phase.

**Drafting Phase.** Drafting methods can be grouped into three categories: *(1) Single-draft.* Early SD methods (Leviathan et al., 2023) inspired PaSS (Monea et al., 2023) and Draft&Verify (Zhang et al., 2024), improving efficiency via multi-token generation or selective layer skipping. GLIDE (Du et al., 2024) (shared KV-cache) offers further speedups but requires task-specific tuning. *(2) Retrieval-based.* LLM-A (Yang et al., 2023) and ReST (He et al., 2023) generate drafts from reference texts, potentially reducing latency, but face database limitations, distribution gaps, and reliance on greedy decoding. *(3) Multi-draft.* Tree-attention frameworks—SpecInfer (Miao et al., 2024), Medusa (Cai et al., 2024), and Eagle (Li et al., 2024; Fan et al., 2026)—expand many branches, quickly exhausting memory. Medusa and Eagle also predict drafts from the target model's hidden features rather than a separate draft model, further boosting speed but requires task-specific tuning.

**Verification Phase.** Verification methods trade fidelity for speed. Lossless approaches (Sun et al., 2023b; Yang et al., 2024; Hu et al., 2025) guarantee exact recovery but are costly. Block Verification (Sun et al., 2024) partially alleviates this bottleneck but offers limited improvement and low interpretability and integrity. Lossy methods—including BiLD (Kim et al., 2023), MTAD (Qin et al.,

2025), DistillSpec (Zhou et al., 2024), Medusa-2 (Cai et al., 2024), SpecCascade (Narasimhan et al., 2024) and CoS (Fu et al., 2025) increase speed but compromise distribution fidelity and require task-specific tuning. In addition, Medusa and EAGLE always accept the first draft token to improve throughput, trading off exact recovery of the target distribution.

## 3    REVISITING TOKENWISE SPECULATIVE DECODING

In tokenwise speculative sampling (Leviathan et al., 2023), each token $x_t$ is drafted from $q(x_t)$ and verified against $p(x_t)$. It is accepted with probability $h(x_t) = \min\{1,\, p(x_t)/q(x_t)\}$, or rejected and replaced from $P_{\text{res}}(x_t)$. Thus the probability that $x_t$ is finally produced ("yielded") is:

$$P(x_t \text{ yielded}) = P(x_t \text{ drafted and accepted}) + P(\tilde{x}_t \text{ drafted and rejected},\ x_t \text{ resampled}). \quad (1)$$

**Accept term.**    If $x_t$ is proposed by $q$ and accepted,

$$P(x_t \text{ drafted and accepted}) = q(x_t)\, h(x_t) = q(x_t)\, \min\{1,\, p(x_t)/q(x_t)\}. \quad (2)$$

**Resampling term.**    When a draft $\tilde{x}_t$ is rejected, the verifier resamples from

$$P_{\text{res}}(x_t) = \frac{p(x_t) - \min\{p(x_t), q(x_t)\}}{\sum_{\tilde{x}_t \in \mathcal{V}} \big(p(\tilde{x}_t) - \min\{p(\tilde{x}_t), q(\tilde{x}_t)\}\big)}.$$

The total probability of rejection is $\sum_{\tilde{x}_t \in \mathcal{V}} q(\tilde{x}_t)(1 - h(\tilde{x}_t))$, giving

$$P(\tilde{x}_t \text{ drafted and rejected},\ x_t \text{ resampled}) = \Big[\sum_{\tilde{x}_t \in \mathcal{V}} q(\tilde{x}_t)(1 - h(\tilde{x}_t))\Big] P_{\text{res}}(x_t). \quad (3)$$

**Final distribution.**    The sum $\sum_{\tilde{x}_t \in \mathcal{V}} q(\tilde{x}_t)\,(1 - h(\tilde{x}_t))$ corresponds to the total *excess mass* assigned by the draft distribution to tokens where it allocates more probability than the target, while the denominator of $P_{\text{res}}(x_t)$ measures the total *deficient mass*, i.e., the probability assigned by the target to tokens where it allocates more than the draft. For tokenwise distributions these match ($D_{\text{LK}}(q,p) = D_{\text{LK}}(p,q)$), so they cancel, yielding

$$P(x_t \text{ is yielded}) = q(x_t)h(x_t) + D_{\text{LK}}(q,p)\frac{p(x_t) - q(x_t)h(x_t)}{D_{\text{LK}}(p,q)} = p(x_t).$$

## 4    THEORETICAL FOUNDATIONS OF HIERARCHICAL SPECULATIVE DECODING

For any **lossless speculative decoding**, the probability of generating an output decomposes into two parts: (1) the probability a draft is *accepted*, becoming the final output, and (2) the probability a draft is *rejected*, triggering a corrective resampling step. In *token-wise* speculative decoding, resampling is straightforward because each token's probability is directly accessible. In contrast, full joint probabilities over sequences are intractable for auto-regressive models. **Hierarchical Speculative Decoding (HSD)** overcomes this via *hierarchical branch resampling*, where multiple resampling distributions at different levels recover *partial target distributions*, which together statistically recover the full distribution. This section formalizes the theoretical foundations.

### 4.1    RECOVERY OF PARTIAL DISTRIBUTIONS

To guide recovery within accessible subsets, we extend the divergence from Leviathan et al. (2023) to partial distributions. Let $\omega$ be a token or sequence, $\Omega$ the full sample space, and $p(\cdot), q(\cdot)$ the target and draft distributions. For $\Omega' \subseteq \Omega$, define the *generalized divergence*:

**Definition 1.  Generalized Divergence.** Given two distributions $p$ and $q$ over a sample space $\Omega$, and a subset $\Omega' \subseteq \Omega$, the *generalized divergence* over $\Omega'$ is defined as:

$$D_{\Omega'}(p,q) = \sum_{\tilde{\omega} \in \Omega'} \max\{p(\tilde{\omega}) - q(\tilde{\omega}),\, 0\}. \quad (4)$$

The *generalized divergence* $D_{\Omega'}(p,q)$ measures the total *deficient mass*, i.e., how much probability mass is missing in the draft $q$ relative to the target $p$ within the subset $\Omega'$. The reverse divergence

$D_{\Omega'}(q, p)$ measures the corresponding *excess mass*. In the whole space $\Omega$, this is symmetric (see Lemma 1 in Section A.1 ) and reduces to the divergence from Leviathan et al. (2023) (see Lemma 2 in Section A.4), which underpins standard token-wise speculative decoding.

Next, we formalize the condition under which the partial target distribution is fully recoverable:

**Theorem 1.** *Partial Distribution Recovery. A target distribution over $\Omega' \subseteq \Omega$ can be fully recovered via resampling iff $D_{\Omega'}(p, q) \leq D_{\Omega'}(q, p)$. (See proof in Section A.2.)*

Intuitively, this ensures the "trigger mass" in the draft is sufficient to compensate for the deficit in the target distribution. Over the full space $\Omega$, symmetry guarantees full recoverability.

## 4.2 Resampling within the Accessible Branch

With these definitions, we analyze resampling within *accessible branches* along a draft sequence. Although computing full joint probabilities is intractable, the probabilities of all next tokens over the vocabulary $\mathcal{V}$ are accessible given any prefix $\boldsymbol{X}_{1:t-1}$. We define a *branch* as:

$$\text{Branch}(\boldsymbol{X}_{1:t-1}) = \{\boldsymbol{X}_{1:t} = (\boldsymbol{X}_{1:t-1}, \tilde{x}_t) \mid \tilde{x}_t \in \mathcal{V}\}. \tag{5}$$

Branch divergence will guide redistribution of excess probability mass to correct local deficits.

Since only joint probabilities $p(\boldsymbol{X}_{1:t})$ within a given branch $\text{Branch}(\boldsymbol{X}_{1:t-1})$ are available, we introduce *branch divergence* to quantify local deficits in the draft:

**Definition 2. Branch Divergence**

$$D_{\text{Branch}}(p, q \mid \boldsymbol{X}_{1:t-1}) = \sum_{\boldsymbol{X}_{1:t} \in \text{Branch}(\boldsymbol{X}_{1:t-1})} \max\{p(\boldsymbol{X}_{1:t}) - q(\boldsymbol{X}_{1:t}), 0\} \tag{6}$$

Branch divergence captures how much probability mass is missing locally. Unlike total divergence, it is inherently asymmetric, motivating the definition of *branch asymmetry*:

**Definition 3. Asymmetry of Branch Divergence**

$$\Delta_{\text{Branch}}(\boldsymbol{X}_{1:t-1}) = D_{\text{Branch}}(p, q \mid \boldsymbol{X}_{1:t-1}) - D_{\text{Branch}}(q, p \mid \boldsymbol{X}_{1:t-1}) \tag{7}$$

Asymmetry essentially reflects the probabilistic imbalance within the current branch. Here, $\Delta_{\text{Branch}} > 0$ indicates a deficit that cannot be corrected within the branch alone, while $\Delta_{\text{Branch}} < 0$ represents excess mass available to support other branches. It can be computed as follows:

**Theorem 2.** *Quantifying Asymmetry of Branch Divergence (see proof in Section A.3):*

$$\Delta_{Branch}(\boldsymbol{X}_{1:t-1}) = p(\boldsymbol{X}_{1:t-1}) - q(\boldsymbol{X}_{1:t-1}), \tag{8}$$

From Theorem 1 and Theorem 2, we conclude that resampling can fully recover the target distribution over a branch whenever the draft has enough probability mass to cover the deficit:

**Corollary 3.** *The target distribution over the $\text{Branch}(\boldsymbol{X}_{1:t-1})$ can be recovered via resampling, under the following condition:*

$$p(\boldsymbol{X}_{1:t-1}) \leq q(\boldsymbol{X}_{1:t-1}) \quad \text{or, equivalently,} \quad r(\boldsymbol{X}_{1:t-1}) \leq 1 \tag{9}$$

*where $r(\boldsymbol{X}_{1:t-1}) = \frac{p(\boldsymbol{X}_{1:t-1})}{q(\boldsymbol{X}_{1:t-1})}$ denotes the probability ratio.*

For drafts of length $\gamma$, the full target distribution cannot be recovered by applying verification solely within the accessible $\text{Branch}(\boldsymbol{X}_{1:\gamma-1})$. However, we observe that the unused probability mass in certain branches can be leveraged to compensate for the unrecoverable mass in other branches, from a statistical perspective. This motivates the hierarchical branch resampling approach discussed next.

## 4.3 Resampling in a Hierarchy of Accessible Branches

Accessible branch divergences naturally form a hierarchical structure that enables systematic redistribution of excess probability mass. Specifically:

**Theorem 4.** *Hierarchy of Branch Divergence*

*The total positive asymmetry of branch divergence across child branches is equal to the parent branch divergence, and vice versa. Specifically:*

$$\sum_{\Delta_{Branch}(\boldsymbol{X}_{1:t-2},\tilde{x}_{t-1})>0} \Delta_{Branch}(\boldsymbol{X}_{1:t-2},\tilde{x}_{t-1}) = D_{Branch}(p,q \mid \boldsymbol{X}_{1:t-2}), \quad \text{and vice versa,} \quad (10)$$

*where $\boldsymbol{X}_{1:t-2}, \tilde{x}_{t-1}$ ranges over all possible Branches with the shared prefix $\boldsymbol{X}_{1:t-2}$, and $\boldsymbol{X}_{1:t-2}$ is the accessible branch along the draft sequence. (See Section A.5 for the proof.)*

This result guarantees that excess mass from overrepresented branches can be aggregated to offset deficits in underrepresented branches. Thus, hierarchical branch resampling guarantees exact recovery of the target distribution, even when individual branches cannot. This provides a rigorous theoretical foundation for deriving Hierarchical Speculative Decoding.

---

**Algorithm 1** Naive HSD

**Require:** Target probabilities: $\{p(\cdot), ..., p(\cdot|\boldsymbol{X}_{1:\gamma})\}$
**Require:** Draft probabilities: $\{q(\cdot), ..., q(\cdot|\boldsymbol{X}_{1:\gamma-1})\}$
**Require:** Draft tokens $\boldsymbol{X}_{1:\gamma} = \{x_1, ..., x_\gamma\}$
1: Initialize $\tau = 0$
2: **for** $t$ **in** $\gamma : 1$ **do**
3:     Sample $\eta_t \sim U(0,1)$
4:     **if** $h_t \geq \eta_t$ **then**
5:         Set $\tau = t$           *#accept $\boldsymbol{X}_{1:t}$*
6:         **break**
7:     **else**
8:         Set $\tau = t - 1$      *#reject $x_t$*
9:         **continue**        *#step back*
10:     **end if**
11: **end for**
12: **if** $\tau = \gamma$ **then**
13:     Sample token from $p(\cdot|\boldsymbol{X}_{1:\gamma})$   *#bonus token*
14: **else**
15:     **for** $t$ **in** $\tau : \gamma - 1$ **do**
16:         Sample token from $P_{\text{res}}(\cdot \mid \boldsymbol{X}_{1:t})$ *#resample*
17:     **end for**
18: **end if**
**Ensure:** $[\boldsymbol{X}_{1:\tau}, \tilde{x}_{\tau+1}, \ldots, \tilde{x}_\gamma]$

**Algorithm 2** HSD

**Require:** Target probabilities: $\{p(\cdot), ..., p(\cdot|\boldsymbol{X}_{1:\gamma})\}$
**Require:** Draft probabilities: $\{q(\cdot), ..., q(\cdot|\boldsymbol{X}_{1:\gamma-1})\}$
**Require:** Draft tokens $\boldsymbol{X}_{1:\gamma} = \{x_1, ..., x_\gamma\}$
1: Initialize $\tau = 0$
2: **for** $t$ **in** $\gamma : 1$ **do**
3:     Sample $\eta_t \sim U(0,1)$
4:     **if** $h_t \geq \eta_t$ **then**
5:         Set $\tau = t$           *#accept $\boldsymbol{X}_{1:t}$*
6:         **break**
7:     **else**
8:         Set $\tau = t - 1$      *#reject $x_t$*
9:         **continue**        *#step back*
10:     **end if**
11: **end for**
12: **if** $\tau = \gamma$ **then**
13:     Sample token from $p(\cdot|\boldsymbol{X}_{1:\gamma})$   *#bonus token*
14: **else**
15:     Sample token from $P_{\text{res}}^*(\cdot \mid \boldsymbol{X}_{1:\tau})$   *#resample*
16: **end if**
**Ensure:** $[\boldsymbol{X}_{1:\tau}, \text{token}]$

---

# 5   HIERARCHICAL SPECULATIVE DECODING

Guided by the theoretical foundations, we first develop a *naive algorithm* (see 5.1) that exactly recovers the target distribution. The procedure evaluates a candidate sequence $\boldsymbol{X}_{1:\gamma}$ and scans backward to identify the longest accepted prefix $\boldsymbol{X}_{1:\tau}$, then recursively resamples positions $\tau + 1$ through $\gamma$ using the corresponding distributions from the resampling hierarchy.

This naive approach, however, still requires $\gamma - \tau + 1$ additional calls to the target model, since the resampled branches are inaccessible. To remove this overhead, we introduce *Capped Branch Resampling*, yielding our final *Hierarchical Speculative Decoding (HSD)*. HSD recovers the target distribution with just one resampling step within the accessible branches. Concretely, after the resampling step at line 15 in Algorithm 2, HSD only needs to sample from the target distribution to continue generation until $\gamma$, which can be replaced by another speculative decoding step, eliminating additional target calls.

## 5.1   NAIVE HIERACHICIAL SPECULATIVE DECODING

Specifically, the acceptance probability is computed according to the following formula:

> **Acceptance Probability** $h_\gamma = \min\{r(\boldsymbol{X}_{1:\gamma}), 1\}$, and when $t < \gamma$:
>
> $$h_t = \frac{D_{\text{Branch}}(p, q \mid \boldsymbol{X}_{1:t})}{\max\{D_{\text{Branch}}(p, q \mid \boldsymbol{X}_{1:t}), D_{\text{Branch}}(q, p \mid \boldsymbol{X}_{1:t})\}}, \tag{11}$$
>
> **Branch Resampling Probability** (line 17 in Algorithm 1):
> $$P_{res}(x_t \mid \boldsymbol{X}_{1:t-1}) = \frac{\max\{p(\boldsymbol{X}_{1:t}) - q(\boldsymbol{X}_{1:t}), 0\}}{D_{\text{Branch}}(p, q \mid \boldsymbol{X}_{1:t-1})} \tag{12}$$

*Branch Divergence* $D_{\text{Branch}}(p, q \mid \boldsymbol{X}_{1:t-1})$ is defined in Definition 2. By construction, the *Branch Resampling Probability* is defined within the accessible Branch($\boldsymbol{X}_{1:t-1}$), i.e., $P_{\text{res}}(\boldsymbol{X}_{1:t} \mid \text{Branch}(\boldsymbol{X}_{1:t-1}))$, which reduces to the token-level form $P_{\text{res}}(x_t \mid \boldsymbol{X}_{1:t-1})$.

The probability of the Target Model generating a sequence $\boldsymbol{X}_{1:\gamma}$ can be decomposed into two disjoint events: (i) full acceptance of the draft, or (ii) at least one rejection followed by resampling:

$$\begin{aligned} P\big(\boldsymbol{X}_{1:\gamma} \text{ is yielded}\big) &= P\big(\boldsymbol{X}_{1:\gamma} \text{ is sampled as draft, } \boldsymbol{X}_{1:\gamma} \text{ is accepted}\big) \\ &+ \sum_{\tilde{\boldsymbol{X}}_{1:\gamma} \neq \boldsymbol{X}_{1:\gamma}} P(\tilde{\boldsymbol{X}}_{1:\gamma} \text{ sampled and rejected, } \boldsymbol{X}_{1:\gamma} \text{ resampled}). \end{aligned} \tag{13}$$

Accept term: probability for the case when $\boldsymbol{X}_{1:\gamma}$ is sampled as draft and then directly accepted.

$$P\big(\boldsymbol{X}_{1:\gamma} \text{ is sampled as draft, } \boldsymbol{X}_{1:\gamma} \text{ is accepted}\big) = \underbrace{q(\boldsymbol{X}_{1:\gamma})}_{\text{sample probability}} \underbrace{\min\{r(\boldsymbol{X}_{1:\gamma}), 1\}}_{\text{accept probability at } \gamma} \tag{14}$$

If $r(\boldsymbol{X}_{1:\gamma}) \leq 1$, this equals to the target probability $p(\boldsymbol{X}_{1:\gamma})$. Otherwise, it is equal to $q(\boldsymbol{X}_{1:\gamma})$, and the residual probability $p(\boldsymbol{X}_{1:\gamma}) - q(\boldsymbol{X}_{1:\gamma})$ is compensated via resampling.

Resampling term (partially resampled): This term accounts for all cases where $\boldsymbol{X}_{1:\gamma}$ is obtained by resampling. Note that the accepted prefix must exactly match the corresponding subsequence of $\boldsymbol{X}_{1:\gamma}$ for this contribution to apply. Therefore, we can further decompose it by summing over all possible positions $\tau + 1$ of the first rejected token, with $\tau$ being the length of the longest accepted prefix:

$$\sum_{\tau=0}^{\gamma} \sum_{\tilde{\boldsymbol{X}}_{\tau+1:\gamma}} P(\tilde{\boldsymbol{X}}_{1:\gamma} \text{ sampled and rejected, } \boldsymbol{X}_{1:\gamma} \text{ resampled}) =$$
$$\sum_{\tau=0}^{\gamma} \sum_{\tilde{\boldsymbol{X}}_{\tau+1:\gamma}} q(\boldsymbol{X}_{1:\tau} \tilde{\boldsymbol{X}}_{\tau+1:\gamma}) \cdot \prod_{t=\tau+1}^{\gamma} (1 - h_t) \quad \cdot h_\tau \boldsymbol{X}_{1:\tau} \cdot \prod_{t=\tau+1}^{\gamma} P_{\text{res}}(x_t) \tag{15}$$

**Explanation of terms:**

1. **Sampling:** $q(\boldsymbol{X}_{1:\tau} \tilde{\boldsymbol{X}}_{\tau+1:\gamma})$ is the probability of generating the initial draft sequence.
2. **Backward Scan:** $\prod_{t=\tau+1}^{\gamma}(1 - h_t)$ corresponds to scanning backward from the end, rejecting tokens until the first accepted prefix is found.
3. **Acceptance:** $h_\tau$ is the probability of accepting the longest prefix $\boldsymbol{X}_{1:\tau}$.
4. **Resampling:** $\prod_{t=\tau+1}^{\gamma} P_{\text{res}}(x_t)$ resamples the remaining positions to recover exactly the target probability.

This decomposition defines the procedure underlying Algorithm 1 and provides the basis for its provable losslessness. The complete proof is given in Section B.2, together with an illustrative example Section B.1 showing how naive HSD recovers the target distribution.

## 5.2 HIERARCHICAL SPECULATIVE DECODING WITH CAPPED BRANCH RESAMPLING

To introduce the capped branch sampling, we first define the *Maximum Prefix Ratio Index*.

**Definition 4. Maximum Prefix Ratio Index** For candidate tokens $\boldsymbol{X}_{1:t}$, the *Maximum Prefix Ratio Index* $m(\boldsymbol{X}_{1:t})$ is the position in the prefix $\boldsymbol{X}_{1:t-1}$ where the joint probability ratio $r(\boldsymbol{X}_{1:i})$ is maximized; if no prefix exceeds 1, we set $m(\boldsymbol{X}_{1:t}) = 0$:

$$m(\boldsymbol{X}_{1:t}) = \arg \max_{1 \leq i < t} r(\boldsymbol{X}_{1:i}) \;\; \text{or} \;\; 0 \text{ if } \max_{1 \leq i < t} r(\boldsymbol{X}_{1:i}) \leq 1.$$

Based on the *Maximum Prefix Ratio Index*, we define the *Capped Prefix Ratio* $r^*$ as follows:

**Definition 5. Capped Prefix Ratio**

$$r^*(\boldsymbol{X}_{1:t}) = \min\{r(\boldsymbol{X}_{1:m(\boldsymbol{X}_{1:t})}), 1\} r(\boldsymbol{X}_{m(\boldsymbol{X}_{1:t})+1:t}). \tag{16}$$

By Definition 5, we have $r(\boldsymbol{X}_{1:m(\boldsymbol{X}_{1:t})}) > 1$, and according to Equation (16), this implies the identity $r^*(\boldsymbol{X}_{1:t}) = r\big(\boldsymbol{X}_{m(\boldsymbol{X}_{1:t})+1:t}\big)$.

Then we define the *Capped Branch Divergence*:

**Definition 6. Capped Branch Divergence**

$$D_{\text{Branch}}^*(p, q \mid \boldsymbol{X}_{1:t-1}) = \sum_{\substack{\boldsymbol{X}_{1:t} \in \text{Branch}(\boldsymbol{X}_{1:t-1}); \\ r^*(\boldsymbol{X}_{1:t}) > 1}} (r^*(\boldsymbol{X}_{1:t}) - 1) \, q(\boldsymbol{X}_{1:t}) \tag{17}$$

$$D_{\text{Branch}}^*(q, p \mid \boldsymbol{X}_{1:t-1}) = \sum_{\substack{\boldsymbol{X}_{1:t} \in \text{Branch}(\boldsymbol{X}_{1:t-1}); \\ r^*(\boldsymbol{X}_{1:t}) \leq 1}} (1 - r^*(\boldsymbol{X}_{1:t})) \, q(\boldsymbol{X}_{1:t}) \tag{18}$$

Finally, the acceptance probability is computed according to the following formula:

> *Acceptance Probability* $h_\gamma = \min\{r^*(\boldsymbol{X}_{1:\gamma}), 1\}$, and when $t < \gamma$:
>
> $$h_t = \frac{D_{\text{Branch}}^*(p, q \mid \boldsymbol{X}_{1:t})}{D_{\text{Branch}}^*(q, p \mid \boldsymbol{X}_{1:t})}, \tag{19}$$
>
> *Capped Branch Resampling Probability* (line 15 in Algorithm 2):
> $$P_{\text{res}}^*(x_t \mid \boldsymbol{X}_{1:t-1}) = \frac{\max\{q(\boldsymbol{X}_{1:t})(r^*(\boldsymbol{X}_{1:t}) - 1), 0\}}{D_{\text{Branch}}^*(p, q \mid \boldsymbol{X}_{1:t-1})} \tag{20}$$

We refer to the above strategy as *Capped Branch Resampling*. It plays a central role in enabling efficient resampling within the hierarchical branch resampling framework. The resampling distribution in Equation (20) enables recovery of the full target distribution with only a single resampling step for branches with negative asymmetry. The remaining positions can then be directly sampled from the target model, aligning with the start of the next speculative decoding step and thus incurring no extra computational cost.

We briefly clarify the core mechanism by which capping preserves the target joint distribution. From Definition 5 and Definition 6, it follows that $D_{\text{Branch}}^*(p, q \mid \boldsymbol{X}_{1:t}) = \sum_{\boldsymbol{X}_{1:t} \in \text{Branch}(\boldsymbol{X}_{1:t-1})} \max\{q(\boldsymbol{X}_{1:m(\boldsymbol{X}_{1:t})}) p(\boldsymbol{X}_{m(\boldsymbol{X}_{1:t})+1:t}) - q(\boldsymbol{X}_{1:t}), 0\}$. Through the acceptance probability and resampling probability at position $t$, we essentially guarantee that the probability of obtaining $\boldsymbol{X}_{1:t}$ is equal to $q(\boldsymbol{X}_{1:m(\boldsymbol{X}_{1:t})}) p(\boldsymbol{X}_{m(\boldsymbol{X}_{1:t})+1:t})$, partially recovering the probability of the fragment $\boldsymbol{X}_{m(\boldsymbol{X}_{1:t})+1:t}$. And the deficient probability mass $p(\boldsymbol{X}_{1:m(\boldsymbol{X}_{1:t})}) - q(X_{1:m(\boldsymbol{X}_{1:t})})$ is statistically recovered from the resampling distributions in higher hierarchies, which corresponds to the fragments $\tilde{\boldsymbol{X}}_{1:m(\boldsymbol{X}_{1:t})}$ of other trajectories. An illustrative example in Section C.1 demonstrates how the algorithm recovers loss over the entire path, with a further explanation of the capped ratio provided in Section C.3.

### 5.3 COMPUTATIONAL EFFICIENCY

The verification stage in HSD adds negligible overhead compared to the savings from reduced target model forward passes. Thanks to parallelized computations across both the vocabulary and draft

---

**GSM8K Example**

```
Eliza's rate per hour for the first 40 hours she works each week is $10. She also receives
an overtime pay of 1.2 times her regular hourly rate. If Eliza worked for 45 hours this week,
how much are her earnings for this week?

To determine Eliza's earnings for the week, we need to calculate both her regular pay and
her overtime pay.

1. **Calculate Regular Pay:**
   - Eliza's regular rate is $10 per hour.
   - She worked 40 hours at her regular rate.
   - Regular pay: 40 hours × $10/hour = $400.

2. **Calculate Overtime Pay:**
   - Eliza worked a total of 45 hours, so
```

---

Figure 2: GSM8K question with the generated prefix. The text shown is the printed output of the decoded string in Markdown format.

positions, HSD is nearly as efficient as tokenwise verification. Runtime measurements (Appendix H) show that verification accounts for less than 1% of total decoding time, with the majority still spent on draft and target forward passes. These results demonstrate that HSD is not only theoretically lossless but also practically efficient, as further confirmed by our experiments in Section 6.

## 5.4 ILLUSTRATIVE EXAMPLE

We use a GSM8K question as a running example to demonstrate HSD (see Figure 2). This example emphasizes the hierarchical acceptance mechanism and the capping behavior that are key to HSD.

**Next-token probabilities.** Under the given prefix, the large (target) and small (draft) models produce the next-token probabilities:

$$\{p(\cdot \mid \mathbf{X}_{1:\gamma})\} = \{0.7156, 1.0000, 1.0000, 1.0000, 0.0000, 0.0000, 0.0000, 1.0000, 0.0000, 0.4968\},$$
$$\{q(\cdot \mid \mathbf{X}_{1:\gamma})\} = \{0.8771, 0.7900, 0.6514, 0.2592, 0.6773, 0.1490, 0.5775, 1.0000, 0.4611, 0.3630\}.$$

The corresponding draft tokens are: $\{\mathtt{she}, \mathtt{work}, \mathtt{ed}, \mathtt{45}, \mathtt{-}, \mathtt{40}, \mathtt{=}, \mathtt{5}, \mathtt{hours}, \mathtt{of}\}$.

**Joint probabilities and ratios.** We compute the joint probabilities along the draft:

$$\{p(\mathbf{X}_{1:t})\}_{t=1}^{\gamma} = \{0.7156, 0.7156, 0.7156, 0.7156, 0, 0, 0, 0, 0, 0\},$$
$$\{q(\mathbf{X}_{1:t})\}_{t=1}^{\gamma} = \{0.8771, 0.6929, 0.4513, 0.1170, 0.0792, 0.0118, 0.0068, 0.0068, 0.0031, 0.0011\},$$
$$\{r(\mathbf{X}_{1:t})\}_{t=1}^{\gamma} = \{0.8159, 1.0327, 1.5855, 6.1171, 0, 0, 0, 0, 0, 0\}.$$

These ratios exhibit early growth above 1 (at $t = 2, 3, 4$) and collapse to 0 once the target probability vanishes (from $t \geq 5$).

**Maximum prefix indices and capped ratios.** Following Definition 4, the maximum prefix indices and capped ratios are $\{m(\mathbf{X}_{1:t})\}_{t=1}^{\gamma} = \{0, 0, 2, 3, 4, 4, 4, 4, 4, 4\}$ and $\{r^*(\mathbf{X}_{1:t})\}_{t=1}^{\gamma} = \{0.8159, 1, 1, 1, 0, 0, 0, 0, 0, 0\}$.

**Capped branch divergences and acceptance.** On the full vocabulary branch $\mathrm{Branch}(\mathbf{X}_{1:t-1})$, we evaluate the capped branch divergences:

$$\{D_{\mathrm{Branch}}^*(p, q \mid \mathbf{X}_{1:t})\}_{t=1}^{\gamma} = \{0.0227, 0.2416, 0.3343, 0.0991, 0, 0, 0, 0, 0, 0\},$$
$$\{D_{\mathrm{Branch}}^*(q, p \mid \mathbf{X}_{1:t})\}_{t=1}^{\gamma} = \{0.1842, 0.2416, 0.3343, 0.0792, 0.0991, 0.0118, 0.0068, 0.0068, 0.0031, 0.0011\}.$$

The hierarchical acceptance (Eq. 19) then yields $\{h_t\}_{t=1}^{\gamma} = \{0.1231, 1, 1, 1, 0, 0, 0, 0, 0, 0\}$.

Acceptance saturates at $t = 2, 3, 4$, implying the first four tokens are validated, i.e., $n_{\mathrm{match}} = 4$.

**Comparison to tokenwise verification.** For a tokenwise baseline that validates strictly left-to-right, the per-position magnitudes are $\{h_t^{\mathrm{tokenwise}}\}_{t=1}^{\gamma} = \{0.8159, 1, 1, 1, 0, 0, 0, 1, 0, 1\}$.

Since the baseline commits at the first position, an initial $h_1 = 0.8159$ may trigger rejection and discard the entire draft block.

## 6 EXPERIMENTS

In this section, we empirically demonstrate the superiority of HSD with comparison on various benchmarks and configurations, comprehensive ablation studies, and in-depth analysis of results.

### 6.1 EXPERIMENT SETTING

**Experiments Setup.** Experiments are conducted with the widely adopted GPTQ-quantized 8-bit instruction-tuned Qwen2.5 series (Bai et al., 2023). By default, we employ the 0.5B as the draft model and 72B as the target models, with a temperature of 1. We leverage GSM8K (Cobbe et al., 2021) for mathematical problem-solving, HumanEval (Chen et al., 2021) for code generation, and CNN/DailyMail (See et al., 2017) for text summarization. All experiments were conducted on a single NVIDIA H20 GPU with 96 GB of memory, unless otherwise specified.

**Baselines and Metrics.** We compare two lossless verification methods—Token-wise and Block-wise—using two metrics: *Block Efficiency* (tokens/step) and *Decoding Speed* (tokens/second). *Block Efficiency* measures the average tokens generated per serial call to the target model, reflecting intrinsic efficiency independent of hardware. *Decoding Speed* indicates tokens produced per second for practical reference. Additional details and extended evaluations are in Section E.

### 6.2 EXPERIMENT RESULTS

**Main results.** Table 1 summarizes the performance of HSD across datasets and model scales using the Qwen2.5 suite (0.5B as draft,14B, 32B, and 72B as targets). Overall, HSD consistently improves both Block Efficiency (BE) and Decoding Speed (DS) relative to Tokenwise and Blockwise verification. For **GSM8K**, the gains are stable across scales, with BE improvements of **5.2%–5.4%** at 14B/32B and **3.3%** at 72B, accompanied by DS increases of up to **10.7%**. On **HumanEval**, the effect is more pronounced: BE rises by **9.5%** and **12.3%** at 14B and 32B, while DS improves by **9.3%** and **11.4%**; even at 72B, HSD maintains positive margins (**3.3%** BE, **4.5%** DS). For **CNN/DailyMail**, the improvements are moderate but consistent, with BE gains of **4.2%–8.4%** and DS gains of **3.4%–7.2%**. On average, HSD provides consistent advantages over Tokenwise and Blockwise verification, with improvements of approximately **6.2% in BE** and **6.7% in DS**.

**Multi-draft.** To demonstrate the compatibility of HSD, we compare it with token-wise verificaiton in a multi-draft setting. For simplicity—and without loss of generality—we adopt Recursive Reject Sampling (RRS) with replacement (Yang et al., 2024) as the baseline for its scalability and independence from complex tree attention mechanisms. Notably, since it is not straightforward to extend blockwise verification to the multi-draft setup, we omit it from our comparison. We evaluated multi-draft

Table 1: Comparison of Block Efficiency (BE) and Decoding Speed (DS) across datasets and model scales. Values in parentheses show percentage improvement over Tokenwise.

| Method | Block Efficiency (Token/Step) | | | Decoding Speed (Token/Second) | | |
|---|---|---|---|---|---|---|
| | 14B | 32B | 72B | 14B | 32B | 72B |
| **GSM8K** | | | | | | |
| Tokenwise | 5.99 | 6.14 | 6.44 | 82.28 | 53.87 | 31.49 |
| Blockwise | 6.13 (+2.3%) | 6.26 (+2.0%) | 6.53 (+1.4%) | 86.06 (+4.6%) | 54.91 (+1.9%) | 31.79 (+1.0%) |
| **HSD (Ours)** | **6.30 (+5.2%)** | **6.47 (+5.4%)** | **6.65 (+3.3%)** | **91.05 (+10.7%)** | **57.12 (+6.0%)** | **32.52 (+3.3%)** |
| **HumanEval** | | | | | | |
| Tokenwise | 4.83 | 4.89 | 5.23 | 74.21 | 45.68 | 26.31 |
| Blockwise | 5.11 (+5.8%) | 5.15 (+5.3%) | 5.34 (+2.1%) | 78.14 (+5.3%) | 48.15 (+5.4%) | 26.96 (+2.5%) |
| **HSD (Ours)** | **5.29 (+9.5%)** | **5.49 (+12.3%)** | **5.40 (+3.3%)** | **81.09 (+9.3%)** | **50.88 (+11.4%)** | **27.48 (+4.4%)** |
| **CNN/DailyMail** | | | | | | |
| Tokenwise | 2.39 | 2.36 | 2.35 | 37.28 | 21.89 | 11.90 |
| Blockwise | 2.50 (+4.6%) | 2.42 (+2.5%) | 2.39 (+1.7%) | 38.54 (+3.4%) | 22.31 (+1.9%) | 12.10 (+1.4%) |
| **HSD (Ours)** | **2.59 (+8.4%)** | **2.46 (+4.2%)** | **2.45 (+4.3%)** | **39.96 (+7.2%)** | **22.78 (+4.1%)** | **12.33 (+3.6%)** |

Table 2: Comparison of our HSD and tokenwise verification in Multi-draft setting.

| Method | Block Efficiency (Token/Step) | | | Decoding Speed (Token/Second) | | |
|---|---|---|---|---|---|---|
| | GSM8K | HumanEval | CNN/DailyMail | GSM8K | HumanEval | CNN/DailyMail |
| Tokenwise | 6.44 | 5.23 | 2.35 | 31.49 | 26.31 | 11.90 |
| **HSD (Ours)** | **6.65 (+3.3%)** | **5.40 (+3.3%)** | **2.45 (+4.3%)** | **32.52 (+3.3%)** | **27.48 (+4.4%)** | **12.33 (+3.6%)** |
| Tokenwise Multi-draft | 8.65 | 7.96 | 3.79 | 37.66 | 35.72 | 15.38 |
| **HSD Multi-draft (Ours)** | **8.89 (+2.8%)** | **8.26 (+3.8%)** | **4.21 (+11.1%)** | **38.41 (+2.0%)** | **36.83 (+3.1%)** | **16.75 (+8.9%)** |

Table 3: Ablations on temperature, draft length, and target model size on GSM8K. Except for the ablation on target model size, we adopt Qwen2.5-0.5B and Qwen2.5-72B as the draft and target pair.

(a) Ablation on **temperature** ($\gamma = 10$).

| Method | Block Efficiency | | | Decoding Speed | | |
|---|---|---|---|---|---|---|
| | $t = 0.6$ | $t = 0.8$ | $t = 1$ | $t = 0.6$ | $t = 0.8$ | $t = 1$ |
| Tokenwise | 6.81 | 6.70 | 6.44 | 32.86 | 32.18 | 31.49 |
| Blockwise | 6.83 | 6.74 | 6.53 | 33.07 | 32.33 | 31.79 |
| Hierarchicial | 6.86 | 6.79 | 6.65 | 33.21 | 32.90 | 32.52 |

(b) Ablation on **draft lengths** ($t = 1$).

| Method | Block Efficiency | | | Decoding Speed | | |
|---|---|---|---|---|---|---|
| | $\gamma = 5$ | $\gamma = 10$ | $\gamma = 15$ | $\gamma = 5$ | $\gamma = 10$ | $\gamma = 15$ |
| Tokenwise | 4.48 | 6.44 | 7.61 | 12.01 | 31.49 | 51.03 |
| Blockwise | 4.52 | 6.53 | 7.74 | 12.14 | 31.79 | 51.75 |
| Hierarchical | 4.59 | 6.65 | 7.88 | 12.35 | 32.52 | 52.95 |

Table 4: Extended experimental results using the LLaMA model family and EAGLE-3 framework on GSM8K. Note that we replace EAGLE-3's tokenwise verification with our HSD, yielding EAGLE-3H.

(a) Evaluation using the LLaMA-3 model family.

| Method | Single-draft | | Multi-draft | |
|---|---|---|---|---|
| | Block Eff. | Decoding Speed | Block Eff. | Decoding Speed |
| Tokenwise | 6.83 | 8.41 | 8.72 | 10.21 |
| Blockwise | 7.32(+7.2%) | 8.87(+5.5%) | N/A | N/A |
| **HSD (Ours)** | **7.43(+8.8%)** | **9.18(+9.2%)** | **9.00(+3.2%)** | **11.02(+7.9%)** |

(b) Integration with EAGLE-3.

| Method | Block Eff. | Decoding Speed |
|---|---|---|
| EAGLE-3 | 3.40 | 71.59 |
| Blockwise | N/A | N/A |
| **EAGLE-3H (Ours)** | **3.55(+4.4%)** | **80.49(+12.4%)** |

generation with 11 candidate drafts in Table 2, and HSD yields an average 5.9% improvement in Block Efficiency and 4.7% improvement in Decoding Speed over token-wise decoding.

**Ablation on Temperature.** We conduct a systematic evaluation of sampling temperature's effect on decoding efficiency, with $t \in \{0.6, 0.8, 1.0\}$ (Table 3(a)). HSD consistently outperforms other approaches across all temperature settings, demonstrating its robustness to temperature variations.

**Ablation on Draft Length.** We evaluate draft lengths $\gamma \in \{5, 10, 15\}$ tokens, where HSD consistently outperforms baselines with increasing efficiency gains (Table 3(b)). At $\gamma = 15$, HSD achieves peak performance with 7.88 tokens/step in block efficiency and 52.95 steps/second in decoding speed, representing improvements of 3.58% and 3.88% over Tokenwise, respectively. The consistent performance advantage across all draft lengths demonstrates HSD's robust scalability.

**Extended Results.** We conducted additional experiments using Llama-3.1-70B-Instruct and Llama-3.1-8B-Instruct pair (non-quantized version), with model weights distributed on 8 H20 GPUs. The results are shown in Table 4a. Moreover, we integrated HSD into the SOTA EAGLE-3-LLaMa3.1-Instruct-8B ($\gamma = 7$) by replacing its tokenwise verifier in Table 4b. Following EAGLE-3, we accept at least the first draft token for a fair comparison. Note that EAGLE-3 utilizes top-K sampling for drafting, making all draft probabilities equal to 1. In this case, any verification method theoretically degenerates into the same behavior and the observed gain in block efficiency of HSD is likely influenced by sampling stochasticity and floating-point precision. However, the observed significant practical speedup in decoding speed is expected, since our implementation (see Section F) avoids the explicit loops in EAGLE's implementation of tokenwise verification.

## 7 CONCLUSION

We present HSD, a lossless verification method that maximizes accepted tokens while provably preserving the full target distribution. Supported by theoretical guarantees and extensive experiments, HSD consistently accelerates inference across models and benchmarks. Its drop-in integration with frameworks like EAGLE-3 demonstrates both practicality and broad applicability. HSD sets a new standard for efficient, lossless speculative decoding in large language models.

ACKNOWLEDGMENTS

This work was supported in part by the University of Washington Faculty Startup Fund, the Carwein–Andrews Fellowship, the UW GSFEI Top Scholar Award, and the U.S. Department of Transportation (USDOT) PacTrans Sub-Center Seed Funding Program. The authors also thank the anonymous reviewers for their insightful comments and constructive feedback.

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

# APPENDIX

## A  THEORETICAL FOUNDATION

### A.1  SYMMETRY OF TOTAL DIVERGENCE

**Lemma 1.** *Symmetry of Total Divergence.*
$$D_\Omega(p, q) = D_\Omega(q, p). \tag{A.1}$$

*Proof.* From Definition 1, we know:

$$
\begin{aligned}
D_\Omega(p, q) - D_\Omega(q, p) &= \sum_{\tilde{\omega} \in \Omega} \max\{p(\tilde{\omega}) - q(\tilde{\omega}), 0\} - \sum_{\tilde{\omega} \in \Omega} \max\{q(\tilde{\omega}) - p(\tilde{\omega}), 0\} \\
&= \sum_{\substack{\tilde{\omega} \in \Omega \\ p(\tilde{\omega}) \geq q(\tilde{\omega})}} (p(\tilde{\omega}) - q(\tilde{\omega})) - \sum_{\substack{\tilde{\omega} \in \Omega \\ q(\tilde{\omega}) > p(\tilde{\omega})}} (q(\tilde{\omega}) - p(\tilde{\omega})) \\
&= \sum_{\tilde{\omega} \in \Omega} p(\tilde{\omega}) - \sum_{\tilde{\omega} \in \Omega} q(\tilde{\omega}) \\
&= 0 \quad \text{(since both } p \text{ and } q \text{ sum to 1 over the full sample space } \Omega\text{)}
\end{aligned}
\tag{A.2}
$$

Thus, $D_\Omega(p, q) = D_\Omega(q, p)$, completing the proof. $\square$

## A.2 PARTIAL DISTRIBUTION RECOVERY

*Proof of Theorem 1.* Let $P(w \text{ is yielded})$ denote the total probability of producing $w \in \Omega'$. By construction, this can be decomposed as

$$P(w \text{ is yielded}) = P(w \text{ is drafted & accepted}) + P(w \text{ is drafted & rejected, } w \text{ is resampled}),$$
$$(A.3)$$

where acceptance occurs with probability $h(w) = \min\{p(w)/q(w), 1\}$, and resampling follows the distribution $P_{\text{res}}(\cdot \mid \Omega')$ with total trigger mass $D_{\Omega'}(q, p)$. Here, the total trigger mass represents the sum of probabilities of all draft outcomes in $\Omega'$ that are rejected. Hence,

$$P(w \text{ is yielded}) = h(w)\, q(w) + D_{\Omega'}(q, p)\, P_{\text{res}}(w \mid \Omega'). \tag{A.4}$$

Noting that $h(w)\, q(w) = \min\{p(w), q(w)\}$, we have

$$P(w \text{ is yielded}) = \min\{p(w), q(w)\} + D_{\Omega'}(q, p)\, P_{\text{res}}(w \mid \Omega'). \tag{A.5}$$

To match the target distribution exactly ($P(w \text{ is yielded}) = p(w)$), we require

$$P_{\text{res}}(w \mid \Omega') = \frac{p(w) - \min\{p(w), q(w)\}}{D_{\Omega'}(q, p)} = \frac{\max\{p(w) - q(w), 0\}}{D_{\Omega'}(q, p)}. \tag{A.6}$$

Summing over all $w \in \Omega'$ gives

$$\sum_{w \in \Omega'} P_{\text{res}}(w \mid \Omega') = \frac{D_{\Omega'}(p, q)}{D_{\Omega'}(q, p)}. \tag{A.7}$$

For $P_{\text{res}}(\cdot \mid \Omega')$ to be a valid probability distribution, this sum must not exceed 1. Therefore, the necessary and sufficient condition is

$$D_{\Omega'}(p, q) \le D_{\Omega'}(q, p), \tag{A.8}$$

which completes the proof. $\qquad \square$

## A.3 QUANTIFICATION ANALYSIS OF ASYMMETRY

*Proof.* From Definition 3 and Definition 2, we obtain:

$$
\begin{aligned}
\Delta_{\text{Branch}}(\boldsymbol{X}_{1:t-1}) &= \sum_{\boldsymbol{X}_{1:t} \in \text{Branch}(\boldsymbol{X}_{1:t-1})} \max\left\{p\left(\boldsymbol{X}_{1:t}\right) - q\left(\boldsymbol{X}_{1:t}\right), 0\right\} \\
&\quad - \sum_{\boldsymbol{X}_{1:t} \in \text{Branch}(\boldsymbol{X}_{1:t-1})} \max\left\{q\left(\boldsymbol{X}_{1:t}\right) - p\left(\boldsymbol{X}_{1:t}\right), 0\right\} \\
&= \sum_{\boldsymbol{X}_{1:t} \in \text{Branch}(\boldsymbol{X}_{1:t-1})} p\left(\boldsymbol{X}_{1:t}\right) - \sum_{\boldsymbol{X}_{1:t} \in \text{Branch}(\boldsymbol{X}_{1:t-1})} q\left(\boldsymbol{X}_{1:t}\right) \\
&= \sum_{x_t \in \mathcal{V}} p\left(\boldsymbol{X}_{1:t-1}\right) p\left(x_t \mid \boldsymbol{X}_{1:t-1}\right) - \sum_{x_t \in \mathcal{V}} q\left(\boldsymbol{X}_{1:t-1}\right) q\left(x_t \mid \boldsymbol{X}_{1:t-1}\right) \\
&= p\left(\boldsymbol{X}_{1:t-1}\right) - q\left(\boldsymbol{X}_{1:t-1}\right) \quad (\text{since } \sum_{x_t \in \mathcal{V}} p(x_t \mid \boldsymbol{X}_{1:t-1}) = 1)
\end{aligned}
\tag{A.9}
$$

$\square$

## A.4 RELATION TO THE DIVERGENCE IN LEVIATHAN ET AL. (2023)

**Lemma 2.** *The total divergence is equivalent to the divergence defined in Leviathan et al. (2023) for token distributions over the full sample space.*

*Proof.* Following Leviathan et al. (2023), let $\tilde{x}$ denote a token, and omit conditions in the token probabilities for simplicity. From Definition 3.2 in Leviathan et al. (2023), we have:

$$
\begin{aligned}
D_{\mathrm{LK}}(p, q) &= \sum_{\tilde{x} \in \Omega} \left| \frac{p(\tilde{x}) - q(\tilde{x})}{2} \right| \\
&= \frac{1}{2} \left( \sum_{\tilde{x} \in \Omega} \max\{p(\tilde{x}) - q(\tilde{x}), 0\} + \sum_{\tilde{x} \in \Omega} \max\{q(\tilde{x}) - p(\tilde{x}), 0\} \right)
\end{aligned}
\tag{A.10}
$$

From Lemma 1, we know that $D_\Omega(p, q) = D_\Omega(q, p)$, so we can write:

$$
\begin{aligned}
D_\Omega(p, q) &= \frac{D_\Omega(p, q) + D_\Omega(q, p)}{2} \\
&= \frac{1}{2} \left( \sum_{\tilde{x} \in \Omega} \max\{p(\tilde{x}) - q(\tilde{x}), 0\} + \sum_{\tilde{x} \in \Omega} \max\{q(\tilde{x}) - p(\tilde{x}), 0\} \right)
\end{aligned}
\tag{A.11}
$$

Therefore, $D_\Omega(p, q) = D_{\mathrm{LK}}(p, q)$, completing the proof. $\square$

## A.5 HIERARCHY OF DIVERGENCE

*Proof.* Proof of Theorem 4.

From Theorem 2, we recall that:

$$
\Delta_{\mathrm{Branch}}(\boldsymbol{X}_{1:t-2}, \tilde{x}_{t-1}) = p(\boldsymbol{X}_{1:t-2}, \tilde{x}_{t-1}) - q(\boldsymbol{X}_{1:t-2}, \tilde{x}_{t-1}).
\tag{A.12}
$$

Therefore, summing over the cases where this difference is positive gives:

$$
\sum_{\Delta_{\mathrm{Branch}}(\boldsymbol{X}_{1:t-2}, \tilde{x}_{t-1}) > 0} \Delta_{\mathrm{Branch}}(\boldsymbol{X}_{1:t-2}, \tilde{x}_{t-1}) = \sum_{\tilde{x}_{t-1} \in \mathcal{V}} \max \left\{ p(\boldsymbol{X}_{1:t-2}, \tilde{x}_{t-1}) - q(\boldsymbol{X}_{1:t-2}, \tilde{x}_{t-1}), 0 \right\}.
\tag{A.13}
$$

By Definition 2, this is precisely the branch divergence one level higher $D_{\mathrm{Branch}}(p, q \mid \boldsymbol{X}_{1:t-2})$, thus completing the proof. $\square$

## B LOSSLESS OF NAIVE HIERARCHICAL SPECULATIVE DECODING

### B.1 ILLUSTRATIVE EXAMPLE

For example, consider the case where $r(\boldsymbol{X}_{1:\gamma}) > 1$, $r(\boldsymbol{X}_{1:\gamma-1}) > 1$, and $r(\boldsymbol{X}_{1:\gamma-2}) \leq 1$. The accept term is simply equal to $q(\boldsymbol{X}_{1:\gamma})$, so we only need to check whether the resampling term equals $p(\boldsymbol{X}_{1:\gamma}) - q(\boldsymbol{X}_{1:\gamma})$. According to Equation (12), we know $P_{\mathrm{res}}(x_{\gamma-2} \mid \boldsymbol{X}_{1:\gamma-1}) = 0$. Consequently, contributions from positions earlier than $\gamma - 1$ in the sum above vanish, which implies that the resampling term for $\boldsymbol{X}_{1:\gamma}$ arises solely from resampling at positions $\gamma$ and $\gamma - 1$ as follows:

$$
\begin{aligned}
&\sum_{\tilde{x}_\gamma} P\big(\text{sample } \boldsymbol{X}_{1:\gamma-1}\tilde{x}_\gamma, \text{reject } \tilde{x}_\gamma, \text{accept } \boldsymbol{X}_{1:\gamma-1}, \text{resample } x_\gamma\big) + \\
&\sum_{\tilde{\boldsymbol{X}}_{\gamma-1:\gamma}} P\big(\text{sample } \boldsymbol{X}_{1:\gamma-2}\tilde{x}_{\gamma-1:\gamma}, \text{reject } \tilde{\boldsymbol{X}}_{\gamma-1:\gamma}, \text{accept } \boldsymbol{X}_{1:\gamma-2}, \text{resample } \boldsymbol{X}_{\gamma-1:\gamma}\big) \\
&= \sum_{\tilde{x}_\gamma} \underbrace{q(\boldsymbol{X}_{1:\gamma-1}\tilde{x}_\gamma)}_{\text{draft probability}} \cdot \underbrace{(1 - h_\gamma)}_{\text{reject backwards at } \tau+1=\gamma} \cdot \underbrace{h_\gamma}_{\text{accept } \boldsymbol{X}_{1:\gamma-1}} \cdot \underbrace{P_{\mathrm{res}}(x_t)}_{\text{resample at } \tau+1=\gamma} + \\
&\quad \sum_{\tilde{x}_{\gamma-1}} \sum_{\tilde{x}_\gamma} \underbrace{q(\boldsymbol{X}_{1:\gamma-2}\tilde{x}_{\gamma-1}\tilde{x}_\gamma)}_{\text{draft probability}} \cdot \underbrace{(1 - h_\gamma)(1 - h_{\gamma-1})}_{\text{reject backwards at } \tau+1=\gamma-1} \cdot \underbrace{h_{\gamma-2}}_{\text{accept } \boldsymbol{X}_{1:\gamma-1}} \cdot \underbrace{P_{\mathrm{res}}(x_{\gamma-1})P_{\mathrm{res}}(x_\gamma)}_{\text{resample at } \tau+1=\gamma}
\end{aligned}
\tag{A.14}
$$

From Definition 2 that the excess probability mass that triggers resampling $D_{\text{Branch}}(q, p \mid \boldsymbol{X}_{1:\gamma-1}) = \sum_{\tilde{x}_\gamma} q(\boldsymbol{X}_{1:\gamma-1}\tilde{x}_\gamma)(1 - h_\gamma)$. Then we have:

$$
\begin{aligned}
= &D_{\text{Branch}}(q, p | \boldsymbol{X}_{1:\gamma-1}) \cdot 1 \cdot \frac{p(\boldsymbol{X}_{1:\gamma}) - q(\boldsymbol{X}_{1:\gamma})}{D_{\text{Branch}}(p, q | \boldsymbol{X}_{1:\gamma-1})} + \\
&\sum_{\tilde{x}_{\gamma-1}} D_{\text{Branch}}(q, p | \boldsymbol{X}_{1:\gamma-2}\tilde{x}_{\gamma-1})(1 - \frac{D_{\text{Branch}}(p, q | \boldsymbol{X}_{1:\gamma-2}\tilde{x}_{\gamma-1})}{D_{\text{Branch}}(q, p | \boldsymbol{X}_{1:\gamma-2}\tilde{x}_{\gamma-1})}) P_{\text{res}}(x_{\gamma-1}) P_{\text{res}}(x_\gamma)
\end{aligned}
\tag{A.15}
$$

From Definition 3 and Theorem 4, we know that $\sum_{\tilde{x}_{\gamma-1}} D_{\text{Branch}}(q, p | \boldsymbol{X}_{1:\gamma-2}\tilde{x}_{\gamma-1}) - D_{\text{Branch}}(p, q | \boldsymbol{X}_{1:\gamma-2}\tilde{x}_{\gamma-1}) = D_{\text{Branch}}(q, p | \boldsymbol{X}_{1:\gamma-2})$. Then we have:

$$
\begin{aligned}
= &\frac{D_{\text{Branch}}(q, p | \boldsymbol{X}_{1:\gamma-1})}{D_{\text{Branch}}(p, q | \boldsymbol{X}_{1:\gamma-1})} \cdot (p(\boldsymbol{X}_{1:\gamma}) - q(\boldsymbol{X}_{1:\gamma})) + \\
&D_{\text{Branch}}(q, p | \boldsymbol{X}_{1:\gamma-2}) \cdot \frac{p(\boldsymbol{X}_{1:\gamma-1}) - q(\boldsymbol{X}_{1:\gamma-1})}{D_{\text{Branch}}(p, q | \boldsymbol{X}_{1:\gamma-2})} \cdot \frac{p(\boldsymbol{X}_{1:\gamma}) - q(\boldsymbol{X}_{1:\gamma})}{D_{\text{Branch}}(p, q | \boldsymbol{X}_{1:\gamma-1})}
\end{aligned}
\tag{A.16}
$$

We know from Definition 3 and Theorem 2 that $p(\boldsymbol{X}_{1:\gamma}) - q(\boldsymbol{X}_{1:\gamma}) = D_{\text{Branch}}(p, q | \boldsymbol{X}_{1:\gamma-1}) - D_{\text{Branch}}(q, p | \boldsymbol{X}_{1:\gamma-1})$. Then we have:

$$
\begin{aligned}
= &\frac{D_{\text{Branch}}(q, p | \boldsymbol{X}_{1:\gamma-1})}{D_{\text{Branch}}(p, q | \boldsymbol{X}_{1:\gamma-1})} \cdot (p(\boldsymbol{X}_{1:\gamma}) - q(\boldsymbol{X}_{1:\gamma})) + \\
&\frac{\big(D_{\text{Branch}}(p, q \mid \boldsymbol{X}_{1:\gamma-1}) - D_{\text{Branch}}(q, p \mid \boldsymbol{X}_{1:\gamma-1})\big)}{D_{\text{Branch}}(p, q \mid \boldsymbol{X}_{1:\gamma-1})} \cdot (p(\boldsymbol{X}_{1:\gamma}) - q(\boldsymbol{X}_{1:\gamma})) \\
= &\, p(\boldsymbol{X}_{1:\gamma}) - q(\boldsymbol{X}_{1:\gamma})
\end{aligned}
\tag{A.17}
$$

$$
\sum_{\tilde{x}_\gamma} P\big(\boldsymbol{X}_{1:\gamma-1}\tilde{x}_\gamma \text{ is sampled}, \tilde{x}_\gamma \text{ is rejected}, \boldsymbol{X}_{1:\gamma-1} \text{ is accepted}, x_\gamma \text{ is resampled}\big) + \tag{A.18}
$$

$$
\sum_{\boldsymbol{X}'_{\gamma-1:\gamma}} P\big(\boldsymbol{X}_{1:\gamma-2}\tilde{\boldsymbol{X}}_{\gamma-1:\gamma} \text{is sampled}, \tilde{\boldsymbol{X}}_{\gamma-1:\gamma}\text{is rejected}, \boldsymbol{X}_{1:\gamma-2}\text{is accepted}, \boldsymbol{X}_{\gamma-1:\gamma}\text{is resampled}\big)
\tag{A.19}
$$

$$
= \sum_{\tilde{x}_\gamma} \underbrace{q(\boldsymbol{X}_{1:\gamma-1}\tilde{x}_\gamma)}_{\text{draft probability}} \cdot \underbrace{(1 - h_\gamma)}_{\text{reject backwards at } \tau+1 = \gamma} \cdot \underbrace{h_\gamma}_{\text{accept } \boldsymbol{X}_{1:\gamma-1}} \cdot \underbrace{P_{\text{res}}(x_t)}_{\text{resample at } \tau+1 = \gamma} + \tag{A.20}
$$

$$
\sum_{\tilde{x}_{\gamma-1}} \sum_{\tilde{x}_\gamma} \underbrace{q(\boldsymbol{X}_{1:\gamma-2}\tilde{x}_{\gamma-1}\tilde{x}_\gamma)}_{\text{draft probability}} \cdot \underbrace{(1 - h_\gamma)(1 - h_{\gamma-1})}_{\text{reject backwards at } \tau+1 = \gamma-1} \cdot \underbrace{h_{\gamma-2}}_{\text{accept } \boldsymbol{X}_{1:\gamma-1}} \cdot \underbrace{P_{\text{res}}(x_{\gamma-1}) P_{\text{res}}(x_\gamma)}_{\text{resample at } \tau+1 = \gamma} \tag{A.21}
$$

From Definition 2 that the excess probability mass that triggers resampling $D_{\text{Branch}}(q, p \mid \boldsymbol{X}_{1:\gamma-1}) = \sum_{\tilde{x}_\gamma} q(\boldsymbol{X}_{1:\gamma-1}\tilde{x}_\gamma)(1 - h_\gamma)$. Then we have

$$
= D_{\text{Branch}}(q, p | \boldsymbol{X}_{1:\gamma-1}) \cdot 1 \cdot \frac{p(\boldsymbol{X}_{1:\gamma}) - q(\boldsymbol{X}_{1:\gamma})}{D_{\text{Branch}}(p, q | \boldsymbol{X}_{1:\gamma-1})} + \tag{A.22}
$$

$$
\sum_{\tilde{x}_{\gamma-1}} D_{\text{Branch}}(q, p | \boldsymbol{X}_{1:\gamma-2}\tilde{x}_{\gamma-1})(1 - \frac{D_{\text{Branch}}(p, q | \boldsymbol{X}_{1:\gamma-2}\tilde{x}_{\gamma-1})}{D_{\text{Branch}}(q, p | \boldsymbol{X}_{1:\gamma-2}\tilde{x}_{\gamma-1})}) P_{\text{res}}(x_{\gamma-1}) P_{\text{res}}(x_\gamma) \tag{A.23}
$$

From Definition 3 and Theorem 4, we know that $\sum_{\tilde{x}_{\gamma-1}} D_{\text{Branch}}(q, p | \boldsymbol{X}_{1:\gamma-2}\tilde{x}_{\gamma-1}) - D_{\text{Branch}}(p, q | \boldsymbol{X}_{1:\gamma-2}\tilde{x}_{\gamma-1}) = D_{\text{Branch}}(q, p | \boldsymbol{X}_{1:\gamma-2})$. Then we have

$$
= \frac{D_{\text{Branch}}(q, p | \boldsymbol{X}_{1:\gamma-1})}{D_{\text{Branch}}(p, q | \boldsymbol{X}_{1:\gamma-1})} \cdot (p(\boldsymbol{X}_{1:\gamma}) - q(\boldsymbol{X}_{1:\gamma})) + \tag{A.24}
$$

$$
D_{\text{Branch}}(q, p | \boldsymbol{X}_{1:\gamma-2}) \cdot \frac{p(\boldsymbol{X}_{1:\gamma-1}) - q(\boldsymbol{X}_{1:\gamma-1})}{D_{\text{Branch}}(p, q | \boldsymbol{X}_{1:\gamma-2})} \cdot \frac{p(\boldsymbol{X}_{1:\gamma}) - q(\boldsymbol{X}_{1:\gamma})}{D_{\text{Branch}}(p, q | \boldsymbol{X}_{1:\gamma-1})} \tag{A.25}
$$

We know from Definition 3 and Theorem 2 that $p(\boldsymbol{X}_{1:\gamma}) - q(\boldsymbol{X}_{1:\gamma} = D_{\text{Branch}}(p, q|\boldsymbol{X}_{1:\gamma-1}) - D_{\text{Branch}}(q, p|\boldsymbol{X}_{1:\gamma-1})$. Then we have

$$= \frac{D_{\text{Branch}}(q, p|\boldsymbol{X}_{1:\gamma-1})}{D_{\text{Branch}}(p, q|\boldsymbol{X}_{1:\gamma-1})} \cdot (p(\boldsymbol{X}_{1:\gamma}) - q(\boldsymbol{X}_{1:\gamma})) + \tag{A.26}$$

$$\frac{\left(D_{\text{Branch}}(p, q \mid \boldsymbol{X}_{1:\gamma-1}) - D_{\text{Branch}}(q, p \mid \boldsymbol{X}_{1:\gamma-1})\right)}{D_{\text{Branch}}(p, q \mid \boldsymbol{X}_{1:\gamma-1})} \cdot (p(\boldsymbol{X}_{1:\gamma}) - q(\boldsymbol{X}_{1:\gamma})) \tag{A.27}$$

$$= p(\boldsymbol{X}_{1:\gamma}) - q(\boldsymbol{X}_{1:\gamma}) \tag{A.28}$$

## B.2 GENERAL PROOF

**Lemma 3** (Rejection-Resampling Sum Reduction (Tokenwise)). *Let $0 < m < \gamma$ be such that the acceptance ratios satisfy:*

$$r(x_\gamma) > 1, \ r(x_{\gamma-1}) > 1, \ \ldots, \ r(x_{\gamma-m+1}) > 1, \quad r(x_{\gamma-m}) \leq 1. \tag{A.29}$$

*Then, the total probability of obtaining the output via resampling over the last $m$ positions is:*

$$\sum_{i=0}^{m-1} P(x_{\gamma-i} \text{ is rejected}) \prod_{j=0}^{i} P(\boldsymbol{X}_{1:\gamma-j} \text{ is resampled}) = p(\boldsymbol{X}_{1:\gamma}) - q(\boldsymbol{X}_{1:\gamma}). \tag{A.30}$$

*Proof.* We begin by defining auxiliary quantities to simplify the notation. For $i = 0, 1, \ldots, m$, let

$$\Delta_i^+ := D_{\text{Branch}}(q, p \mid \boldsymbol{X}_{1:\gamma-i}),$$
$$\Delta_i^- := D_{\text{Branch}}(p, q \mid \boldsymbol{X}_{1:\gamma-i}), \tag{A.31}$$

where $\Delta_i^-$ quantifies the probability mass to be corrected due to overestimation by $q$, and $\Delta_i^+$ represents the mass available to be allocated from alternate paths.

Define also the recursive product term:

$$P_i := \prod_{j=0}^{i} \frac{\Delta_j^+ - \Delta_j^-}{\Delta_{j+1}^+}, \qquad \text{for } 0 \leq i \leq m-1. \tag{A.32}$$

Using these, the rejection-resample contribution becomes:

$$\sum_{i=1}^{m-1} P(x_{\gamma-i} \text{ is rejected}) \prod_{j=0}^{i} P(\boldsymbol{X}_{1:\gamma-j} \text{ is resampled})$$
$$= \sum_{i=1}^{m-1} \Delta_i^- P_i + (\Delta_{m-1}^+ - \Delta_{m-1}^-)P_{m-1}. \tag{A.33}$$

Now observe the recurrence:

$$\Delta_{k+1}^+ P_{k+1} = (\Delta_k^+ - \Delta_k^-)P_k, \tag{A.34}$$

which implies:

$$(\Delta_k^+ - \Delta_k^-)P_k = \Delta_{k+1}^+ P_{k+1}. \tag{A.35}$$

We apply this recurrence in reverse to simplify equation (1) by telescoping the sum:

$$\sum_{i=1}^{m-1} \Delta_i^- P_i + (\Delta_{m-1}^+ - \Delta_{m-1}^-)P_{m-1} = \sum_{i=1}^{m-2} \Delta_i^- P_i + \Delta_{m-1}^+ P_{m-1}$$
$$= \sum_{i=1}^{m-3} \Delta_i^- P_i + \Delta_{m-2}^+ P_{m-2}$$
$$\vdots$$
$$= \Delta_1^+ P_1$$
$$= \Delta_0^+ - \Delta_0^-$$
$$= p(\boldsymbol{X}_{1:\gamma}) - q(\boldsymbol{X}_{1:\gamma}), \tag{A.36}$$

where the final equality follows from the definition:

$$\Delta_0^+ - \Delta_0^- = D_{\text{Branch}}(q, p \mid \boldsymbol{X}_{1:\gamma}) - D_{\text{Branch}}(p, q \mid \boldsymbol{X}_{1:\gamma}) = p(\boldsymbol{X}_{1:\gamma}) - q(\boldsymbol{X}_{1:\gamma}). \quad \text{(A.37)}$$

This completes the proof. □

**Lemma 4** (No Resampling of Earlier Prefixes (Tokenwise)). *Let $\boldsymbol{X}_{1:\gamma} = [x_1, x_2, \ldots, x_\gamma]$ be a token block, and suppose that for some index $m$, the acceptance ratios satisfy:*

$$r(x_\gamma) > 1, \ r(x_{\gamma-1}) > 1, \ \ldots, \ r(x_{\gamma-m+1}) > 1, \quad r(x_{\gamma-m}) \le 1. \quad \text{(A.38)}$$

*Then for all $t \le \gamma - m$, the resampling probability satisfies:*

$$P(\boldsymbol{X}_{1:t} \text{ is resampled}) = 0. \quad \text{(A.39)}$$

*Proof.* We use the resampling probability formula:

$$P_{\text{res}}(\boldsymbol{X}_{1:t}) = \frac{\max\{p(\boldsymbol{X}_{1:t}) - q(\boldsymbol{X}_{1:t}), 0\}}{\max\{D_{\text{Branch}}(p, q \mid \boldsymbol{X}_{1:t}), \ D_{\text{Branch}}(q, p \mid \boldsymbol{X}_{1:t})\}}. \quad \text{(A.40)}$$

At position $t = \gamma - m$, we are given that the acceptance probability

$$r(x_{\gamma-m}) = \min\left\{1, \frac{p(\boldsymbol{X}_{1:\gamma-m})}{q(\boldsymbol{X}_{1:\gamma-m})}\right\} \le 1, \quad \text{(A.41)}$$

implying $p(\boldsymbol{X}_{1:\gamma-m}) < q(\boldsymbol{X}_{1:\gamma-m})$. Therefore,

$$p(\boldsymbol{X}_{1:\gamma-m}) - q(\boldsymbol{X}_{1:\gamma-m}) \le 0, \quad \text{(A.42)}$$

and hence:

$$P_{\text{res}}(\boldsymbol{X}_{1:\gamma-m}) = 0. \quad \text{(A.43)}$$

This completes the proof. □

**Theorem 5** (Lossless).

$$P(\texttt{yield } \boldsymbol{X}_{1:\gamma}) = p(\boldsymbol{X}_{1:\gamma}). \quad \text{(A.44)}$$

*Proof.* The total probability is the sum of the acceptance and resampling paths. We analyze two cases based on the relative probabilities.

**Case 1:** $p(\boldsymbol{X}_{1:\gamma}) < q(\boldsymbol{X}_{1:\gamma})$ In this case, the acceptance probability for the draft is $\frac{p(\boldsymbol{X}_{1:\gamma})}{q(\boldsymbol{X}_{1:\gamma})}$. The probability of generating $\boldsymbol{X}_{1:\gamma}$ via resampling is 0, as there is no probability deficit to recover.

$$\begin{aligned} P(\texttt{yield } \boldsymbol{X}_{1:\gamma}) &= P(\boldsymbol{X}_{1:\gamma} \text{ is accepted}) + P(\boldsymbol{X}_{1:\gamma} \text{ is resampled}) \\ &= q(\boldsymbol{X}_{1:\gamma}) \cdot \frac{p(\boldsymbol{X}_{1:\gamma})}{q(\boldsymbol{X}_{1:\gamma})} + 0 \\ &= p(\boldsymbol{X}_{1:\gamma}). \end{aligned} \quad \text{(A.45)}$$

**Case 2:** $p(\boldsymbol{X}_{1:\gamma}) \ge q(\boldsymbol{X}_{1:\gamma})$ Here, the acceptance probability for the draft is 1. The resampling path must compensate for the probability deficit. Per lemma 3 and lemma 4, the total probability of all relevant resampling paths is exactly $p(\boldsymbol{X}_{1:\gamma}) - q(\boldsymbol{X}_{1:\gamma})$.

$$\begin{aligned} P(\texttt{yield } \boldsymbol{X}_{1:\gamma}) &= P(\boldsymbol{X}_{1:\gamma} \text{ is accepted}) + P(\boldsymbol{X}_{1:\gamma} \text{ is resampled}) \\ &= q(\boldsymbol{X}_{1:\gamma}) \cdot 1 + \big(p(\boldsymbol{X}_{1:\gamma}) - q(\boldsymbol{X}_{1:\gamma})\big) \\ &= p(\boldsymbol{X}_{1:\gamma}). \end{aligned} \quad \text{(A.46)}$$

These two cases cover all probability events. In both cases, the total probability correctly recovers $p(\boldsymbol{X}_{1:\gamma})$, proving the method is lossless.

□

## C  LOSSLESS OF HIERARCHICAL SPECULATIVE DECODING

### C.1  ILLUSTRATIVE EXAMPLE

Let $p(\cdot)$ be the target and $q(\cdot)$ the draft. For a prefix $\boldsymbol{X}_{1:t}$,

$$r(\boldsymbol{X}_{1:t}) \; := \; \frac{p(\boldsymbol{X}_{1:t})}{q(\boldsymbol{X}_{1:t})}, \qquad r(\boldsymbol{X}_{a+1:b} \mid \boldsymbol{X}_{1:a}) \; := \; \frac{p(\boldsymbol{X}_{a+1:b} \mid \boldsymbol{X}_{1:a})}{q(\boldsymbol{X}_{a+1:b} \mid \boldsymbol{X}_{1:a})},$$

so $r(\boldsymbol{X}_{1:b}) = r(\boldsymbol{X}_{1:a}) \, r(\boldsymbol{X}_{a+1:b} \mid \boldsymbol{X}_{1:a})$. Let $m$ be the last (largest) index $< \gamma$ at which the running maximum of $r(\boldsymbol{X}_{1:t})$ is attained and exceeds 1; let $n < m$ be the previous such index (two-peak case).

As definition 4, define the capped ratio at the end of the draft as

$$r^*(\boldsymbol{X}_{1:\gamma}) \; := \; \min\{r(\boldsymbol{X}_{1:m}), 1\} \, r(\boldsymbol{X}_{m+1:\gamma} \mid \boldsymbol{X}_{1:m}) \; = \; r(\boldsymbol{X}_{m+1:\gamma} \mid \boldsymbol{X}_{1:m}) \; \leq 1,$$

and the *accept* term

$$A_\gamma \; := \; q(\boldsymbol{X}_{1:\gamma}) \, r^*(\boldsymbol{X}_{1:\gamma}).$$

We will also use three *resample* contributions: $T_\gamma$ (at level $\gamma$), $T_m$ (at level $m$), and $T_n$ (at level $n$).

**two-peak example:** $n < m < \gamma$ From definition 4, we have $r(\boldsymbol{X}_{1:n}) > 1$, then $r(\boldsymbol{X}_{1:m}) > r(\boldsymbol{X}_{1:n})$, and no larger value occurs in $(m, \gamma)$. This forces $r(\boldsymbol{X}_{n+1:m} \mid \boldsymbol{X}_{1:n}) > 1$; otherwise $m$ could not be a new maximum.

**Step 1: accept + top-level resample** Since $r^*(\boldsymbol{X}_{1:\gamma}) = r(\boldsymbol{X}_{m+1:\gamma} \mid \boldsymbol{X}_{1:m}) \leq 1$,

$$A_\gamma \; = \; q(\boldsymbol{X}_{1:\gamma}) \, r(\boldsymbol{X}_{m+1:\gamma} \mid \boldsymbol{X}_{1:m}) \; = \; q(\boldsymbol{X}_{1:m}) \, p(\boldsymbol{X}_{m+1:\gamma} \mid \boldsymbol{X}_{1:m}), \qquad T_\gamma \; = \; 0,$$

so

$$H_1 \; := \; A_\gamma + T_\gamma \; = \; q(\boldsymbol{X}_{1:m}) \, p(\boldsymbol{X}_{m+1:\gamma} \mid \boldsymbol{X}_{1:m}).$$

*Intuition.* The suffix $\boldsymbol{X}_{m+1:\gamma}$ is now under $p$; the prefix $\boldsymbol{X}_{1:m}$ is still under $q$.

**Step 2: add the $m$-term** Let $R_{n \to m} := r(\boldsymbol{X}_{n+1:m} \mid \boldsymbol{X}_{1:n}) > 1$. The resample at level $m$ contributes

$$T_m \; := \; q(\boldsymbol{X}_{1:m}) \, (R_{n \to m} - 1) \, p(\boldsymbol{X}_{m+1:\gamma} \mid \boldsymbol{X}_{1:m}),$$

hence

$$H_2 \; := \; H_1 + T_m \; = \; R_{n \to m} \, q(\boldsymbol{X}_{1:m}) \, p(\boldsymbol{X}_{m+1:\gamma} \mid \boldsymbol{X}_{1:m}) \; = \; q(\boldsymbol{X}_{1:n}) \, p(\boldsymbol{X}_{n+1:\gamma} \mid \boldsymbol{X}_{1:n}).$$

*Intuition.* The block $\boldsymbol{X}_{n+1:m}$ is converted to $p$; only $\boldsymbol{X}_{1:n}$ remains under $q$.

**Step 3: add the $n$-term** If $r(\boldsymbol{X}_{1:n}) > 1$,

$$T_n \; := \; q(\boldsymbol{X}_{1:n}) \, (r(\boldsymbol{X}_{1:n}) - 1) \, p(\boldsymbol{X}_{n+1:\gamma} \mid \boldsymbol{X}_{1:n}), \qquad H_3 \; := \; H_2 + T_n \; = \; p(\boldsymbol{X}_{1:\gamma}).$$

If instead $r(\boldsymbol{X}_{1:n}) \leq 1$, then $T_n = 0$ and $H_2 = p(\boldsymbol{X}_{1:\gamma})$ already.

**Intuition.**  Each nonzero term "tops up" the exact deficit of $q$ on its block until the whole path is under $p$. Thus

$$\boxed{A_\gamma + T_\gamma + T_m + T_n \; = \; p(\boldsymbol{X}_{1:\gamma})}$$

in this two-peak case, exhibiting the (lossless) invariance of the total probability under the HSD accept–resample rule.

### C.2  GENERAL PROOF

**Definition 7** (Sequence of Unique Capping Indices)**.** For a given maximum sequence length $\gamma$, the sequence of maximum prefix ratio indices $(m(1), m(2), \ldots, m(\gamma))$ is generated according to Definition 4. Let $\mathcal{U}$ be the set of unique values in the sequence of capping indices:

$$\mathcal{U} = \{m(t) \mid 1 < t \leq \gamma\} \tag{A.47}$$

The **Sequence of Unique Capping Indices**, denoted by $M^*$, is the ordered sequence of the elements in $\mathcal{U}$:

$$M^* = (m_1^*, \ldots, m_L^*) \tag{A.48}$$

where $m_1^* < \ldots < m_L^*$ and $L$ is the total number of unique capping points.

With these definitions, we can now establish the key properties of the prefix-capped joint ratio:

**Lemma 5** (Property of $r^*(\boldsymbol{X}_{1:i})$ between neighboring unique capping indices). *Let $m_l^*$ and $m_{l+1}^*$ be two consecutive unique capping indices, and suppose*

$$m_l^* < i < m_{l+1}^*. \tag{A.49}$$

*For every such $i$, we have $r^*(\boldsymbol{X}_{1:i}) \leq 1$.*

**Lemma 6** (Property of $r^*(\boldsymbol{X}_{1:m_l^*})$ at unique capping indices). *Let $m_{l-1}^*$ and $m_l^*$ be two consecutive unique capping indices, we have*

$$r^*(\boldsymbol{X}_{1:m_l^*}) = r(\boldsymbol{X}_{m_{l-1}^*+1:m_l^*}) > 1$$

.

We now define the acceptance and resampling probability masses:

**Definition 8** (Accepted Probability Mass). The probability mass for accepting the full sequence $\boldsymbol{X}_{1:\gamma}$ is:

$$P(\boldsymbol{X}_{1:\gamma} \text{ is accepted}) = \min(1, r^*(\boldsymbol{X}_{1:\gamma}))\, q(\boldsymbol{X}_{1:\gamma}), \tag{A.50}$$

**Definition 9** (Resampling Probability Mass). Let $\boldsymbol{X}_{1:\gamma}$ be a full sequence of length $\gamma$, and let $M^* = (m_1^*, m_2^*, \ldots, m_L^*)$ be its Sequence of Unique Capping Indices. The total probability mass under the draft $q$ and target $p$ of generating this sequence can be decomposed as:

**Total Generation Probability**

$$
\begin{aligned}
P(\boldsymbol{X}_{1:\gamma} \text{ is generated}) &= P(\boldsymbol{X}_{1:\gamma} \text{ is accepted}) + P(\boldsymbol{X}_{1:\gamma} \text{ is resampled}) \\
&= \min(1,\, r^*(\boldsymbol{X}_{1:\gamma}))\, q(\boldsymbol{X}_{1:\gamma}) \\
&\quad + \sum_{l=1}^{L} \max(0,\, r(\boldsymbol{X}_{m_{l-1}^*+1:m_l^*}) - 1)\, q(\boldsymbol{X}_{1:m_l^*})\, p(\boldsymbol{X}_{m_l^*+1:\gamma} \mid \boldsymbol{X}_{1:m_l^*}) \\
&\quad + \max(0,\, r^*(\boldsymbol{X}_{1:\gamma}) - 1)\, q(\boldsymbol{X}_{1:\gamma-1})\, p(x_\gamma \mid \boldsymbol{X}_{1:\gamma-1})
\end{aligned}
\tag{A.51}
$$

We now establish the key lemma that characterizes the resampling probability mass:

**Lemma 7** (Hierarchical Resampling Probability Mass). *The total generation probability can be decomposed into acceptance and resampling masses as stated in Definition 9. Only unique capping indices contribute to resampling mass, and the explicit form for the resampling mass at each unique capping index is:*

$$
\begin{aligned}
&P(\boldsymbol{X}_{1:m_l^*} \text{ is resampled})\, p(\boldsymbol{X}_{m_l^*+1:\gamma} \mid \boldsymbol{X}_{1:m_l^*}) \\
&= \max(0,\, r(\boldsymbol{X}_{m_{l-1}^*+1:m_l^*}) - 1)\, q(\boldsymbol{X}_{1:m_l^*})\, p(\boldsymbol{X}_{m_l^*+1:\gamma} \mid \boldsymbol{X}_{1:m_l^*})
\end{aligned}
\tag{A.52}
$$

To prove the lossless property, we introduce the segmented probability function:

**Definition 10** (Segmented Probability Function). For each $l \in \{1, \ldots, L\}$, we define the segmented probability function $F_l$ as:

$$
\begin{aligned}
F_l &= q(\boldsymbol{X}_{1:m_l^*})\; p(\boldsymbol{X}_{m_l^*+1:\gamma} \mid \boldsymbol{X}_{1:m_l^*}) \\
&= \Big[\prod_{i=1}^{m_l^*} q(x_i \mid \boldsymbol{X}_{1:i-1})\Big]\Big[\prod_{i=m_l^*+1}^{\gamma} p(x_i \mid \boldsymbol{X}_{1:i-1})\Big],
\end{aligned}
\tag{A.53}
$$

This function represents a hybrid probability measure that uses the draft distribution $q$ up to position $m_l^*$ and the target distribution $p$ for the remaining positions, where $\boldsymbol{X}_{1:0}$ is equal to the prefix.

We establish the telescoping property of resampling mass:

**Lemma 8** (Telescoping of Resampling Mass). *For each $l \in \{1, \ldots, L\}$, the mass of the resampling at the unique capping index $m_l^*$ can be expressed as:*

$$P(\boldsymbol{X}_{1:m_l^*} \text{ is resampled}) = F_{l-1} - F_l. \tag{A.54}$$

*Proof.* we need to show that the resampling mass at the unique capping index $m(l)$ equals $F_{l-1} - F_l$.

1. EXPRESS $F_{l-1}$ IN TERMS OF $F_l$.   We have

$$P\big(\boldsymbol{X}_{1:m_l^*} \text{ is resampled}\big) = \big(r(\boldsymbol{X}_{m_{l-1}^*+1:m_l^*}) - 1\big)\, q\big(\boldsymbol{X}_{1:m_l^*}\big)\, p\big(\boldsymbol{X}_{m_l^*+1:\gamma} \mid \boldsymbol{X}_{1:m_l^*}\big)$$

First note

$$q\big(\boldsymbol{X}_{1:m_{l+1}^*}\big) = q\big(\boldsymbol{X}_{1:m_l^*}\big)\, q\big(\boldsymbol{X}_{m_l^*+1:m_{l+1}^*} \mid \boldsymbol{X}_{1:m_l^*}\big),$$

and

$$p\big(\boldsymbol{X}_{m_l^*+1:m_{l+1}^*} \mid \boldsymbol{X}_{1:m_l^*}\big) = r\big(\boldsymbol{X}_{m_l^*+1:m_{l+1}^*}\big)\, q\big(\boldsymbol{X}_{m_l^*+1:m_{l+1}^*} \mid \boldsymbol{X}_{1:m_l^*}\big).$$

Hence

$$
\begin{aligned}
F_{l-1} &= q\big(\boldsymbol{X}_{1:m_l^*}\big)\, p\big(\boldsymbol{X}_{m_l^*+1:\gamma} \mid \boldsymbol{X}_{1:m_l^*}\big) \\[4pt]
&= q\big(\boldsymbol{X}_{1:m_l^*}\big)\, p\big(\boldsymbol{X}_{m_l^*+1:m_{l+1}^*} \mid \boldsymbol{X}_{1:m_l^*}\big)\, p\big(\boldsymbol{X}_{m_{l+1}^*+1:\gamma} \mid \boldsymbol{X}_{1:m_{l+1}^*}\big) \\[4pt]
&= q\big(\boldsymbol{X}_{1:m_l^*}\big)\, \Big[r(\boldsymbol{X}_{m_l^*+1:m_{l+1}^*})\, q(\boldsymbol{X}_{m_l^*+1:m_{l+1}^*} \mid \boldsymbol{X}_{1:m_l^*})\Big]\, p\big(\boldsymbol{X}_{m_{l+1}^*+1:\gamma} \mid \boldsymbol{X}_{1:m_{l+1}^*}\big) \\[4pt]
&= r\big(\boldsymbol{X}_{m_l^*+1:m_{l+1}^*}\big)\, \Big[q(\boldsymbol{X}_{1:m_l^*})\, q(\boldsymbol{X}_{m_l^*+1:m_{l+1}^*} \mid \boldsymbol{X}_{1:m_l^*})\Big]\, p\big(\boldsymbol{X}_{m_{l+1}^*+1:\gamma} \mid \boldsymbol{X}_{1:m_{l+1}^*}\big) \\[4pt]
&= r\big(\boldsymbol{X}_{m_l^*+1:m_{l+1}^*}\big)\, q\big(\boldsymbol{X}_{1:m_{l+1}^*}\big)\, p\big(\boldsymbol{X}_{m_{l+1}^*+1:\gamma} \mid \boldsymbol{X}_{1:m_{l+1}^*}\big) \\[4pt]
&= r\big(\boldsymbol{X}_{m_l^*+1:m_{l+1}^*}\big)\, F_l.
\end{aligned}
$$

2. COMPUTE THE DIFFERENCE $F_{l-1} - F_l$.

$$
\begin{aligned}
F_{l-1} - F_l &= \Big[r(\boldsymbol{X}_{m_l^*+1:m_{l+1}^*})\, F_l\Big] - F_l \\[4pt]
&= \big(r(\boldsymbol{X}_{m_l^*+1:m_{l+1}^*}) - 1\big)\, F_l \\[4pt]
&= \big(r(\boldsymbol{X}_{m_l^*+1:m_{l+1}^*}) - 1\big)\, q\big(\boldsymbol{X}_{1:m_{l+1}^*}\big)\, p\big(\boldsymbol{X}_{m_{l+1}^*+1:\gamma} \mid \boldsymbol{X}_{1:m_{l+1}^*}\big) \\[4pt]
&= \big(r(\boldsymbol{X}_{m_{l-1}^*+1:m_l^*}) - 1\big)\, q\big(\boldsymbol{X}_{1:m_l^*}\big)\, p\big(\boldsymbol{X}_{m_l^*+1:\gamma} \mid \boldsymbol{X}_{1:m_l^*}\big).
\end{aligned}
$$

This completes the proof that the resampling mass at segment $l$ equals $F_{l-1} - F_l$.  $\square$

**Theorem 6** (Lossless Recovery). *Under the prefix-adaptive speculative decoding scheme, the total probability of generating any sequence $\boldsymbol{X}_{1:\gamma}$ equals the target distribution probability:*

$$P\big(\boldsymbol{X}_{1:\gamma} \text{ is generated}\big) = p(\boldsymbol{X}_{1:\gamma}). \tag{A.55}$$

*Proof.* From Lemma 7, we have the total generation probability decomposition:

$$
\begin{aligned}
P\big(\boldsymbol{X}_{1:\gamma} \text{ is generated}\big) &= P\big(\boldsymbol{X}_{1:\gamma} \text{ is accepted}\big) + P\big(\boldsymbol{X}_{1:\gamma} \text{ is resampled}\big) + P\big(x_\gamma \text{ is resampled}\big) \\[4pt]
&= \min\big(1,\, r^*(\boldsymbol{X}_{1:\gamma})\big)\, q(\boldsymbol{X}_{1:\gamma}) \\[4pt]
&\quad + \sum_{l=1}^{L} \max\big(0,\, r(\boldsymbol{X}_{m_{l-1}^*+1:m_l^*}) - 1\big)\, q(\boldsymbol{X}_{1:m_l^*})\, p(\boldsymbol{X}_{m_l^*+1:\gamma} \mid \boldsymbol{X}_{1:m_l^*}) \\[4pt]
&\quad + \max\big(0,\, r^*(\boldsymbol{X}_{1:\gamma}) - 1\big)\, q(\boldsymbol{X}_{1:\gamma-1})\, p(x_\gamma \mid \boldsymbol{X}_{1:\gamma-1})
\end{aligned}
\tag{A.56}
$$

From Lemma 8, we know that for each $l \in \{1, \ldots, L\}$:

$$F_{l-1} - F_l = \big(r(\boldsymbol{X}_{m_{l-1}^*+1:m_l^*}) - 1\big)\, q(\boldsymbol{X}_{1:m_l^*})\, p(\boldsymbol{X}_{m_l^*+1:\gamma} \mid \boldsymbol{X}_{1:m_l^*}) \tag{A.57}$$

Therefore, we can rewrite the generation probability as:

$$
\begin{aligned}
P\big(\boldsymbol{X}_{1:\gamma} \text{ is generated}\big) &= \min\big(1,\, r^*(\boldsymbol{X}_{1:\gamma})\big)\, q(\boldsymbol{X}_{1:\gamma}) \\[4pt]
&\quad + \sum_{l=1}^{L} (F_{l-1} - F_l) \\[4pt]
&\quad + \max\big(0,\, r^*(\boldsymbol{X}_{1:\gamma}) - 1\big)\, q(\boldsymbol{X}_{1:\gamma-1})\, p(x_\gamma \mid \boldsymbol{X}_{1:\gamma-1})
\end{aligned}
\tag{A.58}
$$

Since $r^*(\boldsymbol{X}_{1:\gamma}) = \min\{r(\boldsymbol{X}_{1:m_L^*}), 1\} r(\boldsymbol{X}_{m_L^*+1:\gamma})$ and $r(\boldsymbol{X}_{1:m_L^*}) > 1$, we have $r^*(\boldsymbol{X}_{1:\gamma}) = r(\boldsymbol{X}_{m_L^*+1:\gamma})$.

Case 1: If $r(\boldsymbol{X}_{m_L^*+1:\gamma}) \le 1$, then:

$$
\begin{aligned}
&\min\big(1, r^*(\boldsymbol{X}_{1:\gamma})\big) \, q(\boldsymbol{X}_{1:\gamma}) + \max\big(0, r^*(\boldsymbol{X}_{1:\gamma}) - 1\big) \, q(\boldsymbol{X}_{1:\gamma-1}) \, p(x_\gamma \mid \boldsymbol{X}_{1:\gamma-1}) \\
&= r(\boldsymbol{X}_{m_L^*+1:\gamma}) \, q(\boldsymbol{X}_{1:\gamma}) + 0 \\
&= r(\boldsymbol{X}_{m_L^*+1:\gamma}) \, q(\boldsymbol{X}_{1:\gamma}) \\
&= q(\boldsymbol{X}_{1:m_L^*}) \, p(\boldsymbol{X}_{m_L^*+1:\gamma} \mid \boldsymbol{X}_{1:m_L^*}) \\
&= F_L
\end{aligned}
\tag{A.59}
$$

Case 2: If $r(\boldsymbol{X}_{m_L^*+1:\gamma}) > 1$, then there would be another unique capping index beyond $m_L^*$, contradicting the definition of $m_L^*$ as the last unique capping index. Therefore, we must have $r(\boldsymbol{X}_{m_L^*+1:\gamma}) \le 1$, and thus:

$$
\min\big(1, r^*(\boldsymbol{X}_{1:\gamma})\big) \, q(\boldsymbol{X}_{1:\gamma}) + \max\big(0, r^*(\boldsymbol{X}_{1:\gamma}) - 1\big) \, q(\boldsymbol{X}_{1:\gamma-1}) \, p(x_\gamma \mid \boldsymbol{X}_{1:\gamma-1}) = F_L \tag{A.60}
$$

Therefore, we have:

$$
\begin{aligned}
P\big(\boldsymbol{X}_{1:\gamma} \text{ is generated}\big) &= F_L + \sum_{l=1}^{L} (F_{l-1} - F_l) \\
&= F_L + (F_0 - F_1) + (F_1 - F_2) + \cdots + (F_{L-1} - F_L) \\
&= F_L + F_0 - F_L \\
&= F_0
\end{aligned}
\tag{A.61}
$$

Now we evaluate $F_0$. From Definition 10, we have:

$$
F_0 = q(\boldsymbol{X}_{1:m_0^*}) \, p(\boldsymbol{X}_{m_0^*+1:\gamma} \mid \boldsymbol{X}_{1:m_0^*}) \tag{A.62}
$$

By our convention, $m_0^* = 0$, so:

$$
F_0 = q(\boldsymbol{X}_{1:0}) \, p(\boldsymbol{X}_{1:\gamma} \mid \boldsymbol{X}_{1:0}) = 1 \cdot p(\boldsymbol{X}_{1:\gamma}) = p(\boldsymbol{X}_{1:\gamma}) \tag{A.63}
$$

Therefore:

$$
\boxed{P\big(\boldsymbol{X}_{1:\gamma} \text{ is generated}\big) = p(\boldsymbol{X}_{1:\gamma})} \tag{A.64}
$$

This completes the proof of lossless recovery. $\qquad\square$

## C.3 A EXTENDED EXPLAINATION OF CAPPED RATIO

Let $r(x_1), r(x_2 \mid x_1), \ldots, r(x_t \mid \boldsymbol{X}_{1:t-1}) \in \mathbb{R}_{>0}$ be a sequence of ratios.

Define the cumulative product up to index $t$ as:

$$
r(\boldsymbol{X}_{1:t}) = \prod_{i=1}^{t} r(x_i \mid \boldsymbol{X}_{1:i-1}), \tag{A.65}
$$

where $\boldsymbol{X}_{1:0}$ is equal to the prefix.

Let $j^*$ be the last index (up to $k$) such that:

$$
j^* = \max\left\{ j \le k \;\middle|\; r(x_j \mid \boldsymbol{X}_{1:j-1}) > 1 \text{ and } \prod_{i=1}^{j} r(x_i \mid \boldsymbol{X}_{1:i-1}) > 1 \right\} \tag{A.66}
$$

Then the capped cumulative product $\tilde{R}_k$ is given by:

$$
r*(\boldsymbol{X}_{1:t}) = \left( \prod_{i=1}^{j^*} r(x_i \mid \boldsymbol{X}_{1:i-1}) \right) \cdot \left( \prod_{i=j^*+1}^{k} r(x_i \mid \boldsymbol{X}_{1:i-1}) \right) \tag{A.67}
$$

This ensures that the cumulative product is capped at the last index $j^*$ such that the individual ratio $r(x_{j^*}|\boldsymbol{X}_{1:j^*-1}) > 1$ and the cumulative product up to that point also exceeds 1.

When $\gamma$ is 3, lets show simplest example to show the recovery of target probability.

$$P\left(\boldsymbol{X}_{1:3} \text{ is accepted }\right) = q(\boldsymbol{X}_{1:3}) \tag{A.68}$$

$$
\begin{aligned}
P\left(\boldsymbol{X}_{1:3} \text{ is resampled}\right) &= \sum_{i=0}^{\gamma=3} P\left(x_\gamma, x_{\gamma-1}, \ldots, x_{\gamma-i} \text{ are resampled } \mid \boldsymbol{X}_{\gamma-i+1}\right) \\
&= D^*_{\text{Branch}}\left(q, p \mid \boldsymbol{X}_{1:3}\right) \cdot \frac{\max((r(x_3)-1)q(\boldsymbol{X}_{1:3}), 0)}{D^*_{\text{Branch}}\left(q, p \mid \boldsymbol{X}_{1:3}\right)} \\
&+ D^*_{\text{Branch}}\left(q, p \mid \boldsymbol{X}_{1:2}\right) \cdot \frac{\max((r(x_2)-1)q(\boldsymbol{X}_{1:2}), 0)}{D^*_{\text{Branch}}\left(q, p \mid \boldsymbol{X}_{1:2}\right)} \cdot p(x_3|\boldsymbol{X}_{1:2}) \\
&+ D^*_{\text{Branch}}\left(q, p \mid x_1\right) \cdot \frac{\max((r(x_1)-1)q(x_1), 0)}{D^*_{\text{Branch}}\left(q, p \mid x_1\right)} \cdot p(x_3|\boldsymbol{X}_{1:2})p(x_2|x_1)
\end{aligned}
\tag{A.69}
$$

Let's take $\gamma = 3$ as an example, only if $r(\boldsymbol{X}_{1:3}) > 1$, the resampled portion of probability mass is needed. Suppose $r(\boldsymbol{X}_{1:2}) > 1$ with $r(x_1) > 1$ and $r(x_2) < 1$:

$$
\begin{aligned}
&= p(x_3|\boldsymbol{X}_{1:2})p(x_2|x_1)q(x_1) - q(\boldsymbol{X}_{1:3}) + 0 \\
&+ p(x_1)p(x_2|x_1)p(x_3|\boldsymbol{X}_{1:2}) - q(x_1)p(x_2|x_1)p(x_3|\boldsymbol{X}_{1:2}) \\
&= p(\boldsymbol{X}_{1:3}) - q(\boldsymbol{X}_{1:3})
\end{aligned}
\tag{A.70}
$$

## D  EXPECTED NUMBER OF ACCEPTED TOKENS

We conduct efficiency analysis based on the expected acceptance length $\mathbb{E}[\tau]$. For a given draft length $\gamma$, the expected number of accepted tokens for the tokenwise speculative decoding Leviathan et al. (2023), blockwise verification Sun et al. (2024), and our HSD are as follows:

**Lemma 9.** *Expected Number of Accepted Tokens (See Section D.1 for proof.)*

$$\mathbb{E}[\tau]_{token} = \sum_{i=1}^{\gamma} \prod_{k=1}^{i} h_k^{token}, \mathbb{E}[\tau]_{block} = \sum_{i=1}^{\gamma}\left[1 - \prod_{k=i}^{\gamma}\left(1 - h_k^{block}\right)\right], \mathbb{E}[\tau]_{branch} = \sum_{i=1}^{\gamma}\left[1 - \prod_{k=i}^{\gamma}\left(1 - h_k\right)\right] \tag{A.71}$$

We establish Theorem 7, which guarantees that HSD is more efficient than other lossless methods:

**Theorem 7.** *HSD and Blockwise Achieves Better Expected Number of Accepted Tokens*

$$\mathbb{E}[\boldsymbol{\tau}]_{branch} \geq \mathbb{E}[\boldsymbol{\tau}]_{block} \geq \mathbb{E}[\boldsymbol{\tau}]_{token} \tag{A.72}$$

*where equality holds in both inequalities if and only if $\gamma = 1$. (See Section D.2 for proof.)*

We reveal that limitations on acceptance probability in each method directly cause the gap from the ideal case w.r.t. expected accepted tokens. Let $r(x_t) = \frac{p(x_t)}{q(x_t)}$. The acceptance probability of the entire draft $h_\gamma$ is ideally $\min\left\{\prod_{t=1}^{\gamma} r(x_t), 1\right\}$. In contrast, tokenwise acceptance is $h_{token} = \prod_{t=1}^{\gamma} \min\{r(x_t), 1\}$, blockwise adopts $h_{block} = \min\{1, r_\gamma, r_{\gamma-1}r_\gamma, \ldots, r_1 r_2 \cdots r_\gamma\}$ (see Lemma 11), and HSD uses $h_{ours} = \min\left\{\min\left\{\prod_{t=1}^{m(x_\gamma)} r(x_t), 1\right\} \prod_{t=m(x_\gamma)+1}^{\gamma} r(x_t), 1\right\}$. See the average acceptance probability $h_\gamma$ on GSM8K in Fig. A.1.

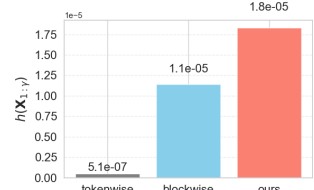

Figure A.1: The average acceptance probability of the entire draft ($\tau = \gamma$) on GSM8K.

Let $\tau \in \{0, 1, \ldots, \gamma\}$ denote the number of accepted tokens in a decoding attempt. Since $\tau$ is a non-negative, integer-valued random variable, the tail-sum identity applies with lattice spacing $a = 1$.

**Lemma 10** (Tail Expectation). *Let $X$ be a non–negative random variable with values in $\{na : n = 0, 1, 2, \dots\}$ for some $a > 0$. Then:*

$$\mathbb{E}[X] = a \sum_{k=1}^{\infty} \Pr(X \geq k). \tag{A.73}$$

*Proof.* Start with the right-hand side:

$$\begin{aligned} a \sum_{k=1}^{\infty} \Pr(X \geq ka) &= a \sum_{k=1}^{\infty} \sum_{\ell \geq k} \Pr(X = \ell a) \\ &= a \sum_{\ell=1}^{\infty} \Pr(X = \ell a) \sum_{k=1}^{\ell} 1 \\ &= \sum_{\ell=1}^{\infty} \ell a \cdot \Pr(X = \ell a) = \mathbb{E}[X]. \end{aligned} \tag{A.74}$$

$\square$

### D.1 EXPECTED TOKEN LENGTH DERIVATION

TOKEN WISE SPECULATIVE DECODING

Referring to *Block-wise Verification* Sun et al. (2024), the authors prove that it achieves a longer expected token length than the token-wise verification Leviathan et al. (2023) (see Appendix B.2 in Sun et al. (2024)).

HIERARCHICAL SPECULATIVE DECODING

Let $\eta_1, \dots, \eta_\gamma \sim \mathcal{U}(0, 1)$ be the random draws used in verification. The accepted length is defined as:

$$\tau := \max \{i \leq \gamma : \eta_i \leq h_i\}, \tag{A.75}$$

where $h_i$ is the acceptance probability at step $i$. By the tail-sum identity:

$$\mathbb{E}[\tau] = \sum_{i=1}^{\gamma} \Pr(\tau \geq i). \tag{A.76}$$

If we define the event $S_i := \{\eta_i \leq h_i\}$, and assume independence of the draws, then:

$$\Pr(\tau \geq i) = 1 - \prod_{k=i}^{\gamma} (1 - h_k). \tag{A.77}$$

Substituting into Equation (A.76), we obtain:

$$\mathbb{E}[\tau] = \sum_{i=1}^{\gamma} \left[ 1 - \prod_{k=i}^{\gamma} (1 - h_k) \right]. \tag{A.78}$$

BLOCKWISE VERIFICATION

In Algorithm 2 (blockwise decoding), the decoding continues even if some $\eta_i > h_i^{\text{block}}$; the resampling happens only at the end. Therefore, the token count $\tau$ still satisfies the same form.

Let $h_i^{\text{block}}$ be the acceptance probability at step $i$ computed via blockwise rules, and define events:

$$S_i := \{\eta_i \leq h_i^{\text{block}}\}, \quad \text{so } \Pr(\overline{S_i}) = 1 - h_i^{\text{block}}. \tag{A.79}$$

We then have:

$$\Pr(\tau \geq i) = 1 - \prod_{k=i}^{\gamma} (1 - h_k^{\text{block}}), \tag{A.80}$$

and hence the expected number of accepted tokens under blockwise decoding is:

$$\mathbb{E}[\tau]_{\text{block}} = \sum_{i=1}^{\gamma} \left[ 1 - \prod_{k=i}^{\gamma} (1 - h_k^{\text{block}}) \right] \tag{A.81}$$

## D.2 TOKEN LENGTH COMPARISON

We re-express the acceptance probability to compare token length between block-wise speculative decoding and our method ( Equation (19)). This yields a more precise comparison via the directional divergence expressions Equation (17) and  Equation (18).

**Capped Branch Divergence Difference**  The difference of capped branch divergence is calculated as:

$$D_{\text{Branch}}^*(p, q \mid \boldsymbol{X}_{1:t}) - D_{\text{Branch}}^*(q, p \mid \boldsymbol{X}_{1:t})$$

$$= \sum_{x_{t+1}} (r^*(\boldsymbol{X}_{1:t+1}) - 1) q(\boldsymbol{X}_{1:t})$$

$$= \sum_{x_{t+1}} \left( \min\{r(\boldsymbol{X}_{0:m(t+1)}), 1\} r(\boldsymbol{X}_{m(t+1)+1:t+1}) - 1 \right) q(\boldsymbol{X}_{0:m(t+1)}) q(\boldsymbol{X}_{m(t+1)+1:t+1})$$

$$= \sum_{x_{t+1}} \left( r(\boldsymbol{X}_{m(\boldsymbol{X}_{1:t+1})+1:t+1}) - 1 \right) q(\boldsymbol{X}_{1:t+1}) \tag{A.82}$$

**Branch Acceptance Probability**  Combine equations (A.82), the acceptance ratio of hierarchical speculative decoding is:

$$h_t^{\text{branch}} = \frac{D_{\text{Branch}}^*(p, q \mid \boldsymbol{X}_{1:t})}{D_{\text{Branch}}^*(q, p \mid \boldsymbol{X}_{1:t})}$$

$$= \frac{D_{\text{Branch}}^*(p, q \mid \boldsymbol{X}_{1:t})}{D_{\text{Branch}}^*(p, q \mid \boldsymbol{X}_{1:t}) + \sum (1 - r(\boldsymbol{X}_{m(\boldsymbol{X}_{1:t+1})+1:t+1})) q(\boldsymbol{X}_{1:t+1})} \tag{A.83}$$

$$= \frac{\sum [r(\boldsymbol{X}_{m(\boldsymbol{X}_{1:t+1})+1:t+1}) - 1]_+}{\sum [r(\boldsymbol{X}_{m(\boldsymbol{X}_{1:t+1})+1:t+1}) - 1]_+ + \sum (1 - r(\boldsymbol{X}_{m(\boldsymbol{X}_{1:t+1})+1:t+1}))}$$

where $[a]_+$ is equal to $\max\{a, 0\}$

**Blockwise Acceptance Ratio**  Algorithm 2 (blockwise decoding), blockwise keeps an internal clamp $p_t = \min\{p_{t-1} r(x_t | \boldsymbol{X}_{1:t-1}), 1\}$, which could be simplified based on Suffix–minimum characterization of $p_t$

**Lemma 11** (Suffix–minimum characterization of $p_t$). *Let $\{r_i\}_{i=1}^{\infty} \subseteq [0, \infty)$ and define the sequence $\{p_t\}_{t \geq 0}$ recursively by*

$$p_0 = 1, \qquad p_t = \min\{ p_{t-1} r_t, 1 \}, \quad t \geq 1. \tag{A.84}$$

*Then for every $t \geq 0$*

$$p_t = \min_{0 \leq s \leq t} \prod_{i=s+1}^{t} r_i, \qquad (\textit{with the empty product for } s = t \textit{ equal to } 1). \tag{A.85}$$

*Equivalently,*

$$p_t = \min\{ 1, r_t, r_{t-1}r_t, \ldots, r_1 r_2 \cdots r_t \}. \tag{A.86}$$

*Proof.* We prove (A.85) by induction on $t$.

**Base case** ($t = 0$). For $t = 0$ the right–hand side becomes

$$\min_{0 \leq s \leq 0}(\text{empty product}) = 1 = p_0, \tag{A.87}$$

so the claim holds.

**Inductive step.** Assume (A.85) holds for some $t - 1 \geq 0$. Using the recurrence,

$$p_t \;=\; \min\{1, \; p_{t-1} \, r_t\}. \tag{A.88}$$

By the induction hypothesis,

$$p_{t-1} = \min_{0 \leq s \leq t-1} \prod_{i=s+1}^{t-1} r_i. \tag{A.89}$$

Substituting,

$$p_t = \min\left\{1, \; \Big[\min_{0 \leq s \leq t-1} \prod_{i=s+1}^{t-1} r_i\Big] r_t\right\}. \tag{A.90}$$

Multiplying every candidate product in the inner minimum by $r_t$ and then taking the outer minimum yields exactly all suffix products

$$\prod_{i=s+1}^{t} r_i \tag{A.91}$$

for $s = 0, \ldots, t - 1$, together with the empty product $1$ for $s = t$. Hence (A.85) holds for $t$, completing the induction. And obviously, $p_t < r(\boldsymbol{X}_{start:t})$, where $start \in (1, t - 1)$ $\qquad\square$

$$
\begin{aligned}
h_t^{\text{block}} &= \frac{\displaystyle\sum_{x_{t+1}}(p_t r(x_{t+1}|\boldsymbol{X}_{1:t}) - 1)_+ \, q(x_{t+1} \mid \boldsymbol{X}_{1:t})}{\displaystyle\sum_{x_{t+1}}(p_t r(x_{t+1}|\boldsymbol{X}_{1:t}) - 1)_+ \, q(x_{t+1} \mid \boldsymbol{X}_{1:t}) \;+\; 1 - p_t} \\[2em]
&= \frac{\displaystyle\sum_{x_{t+1}}(\min\{r(x_{t+1}), r(\boldsymbol{X}_{t:t+1}), r(\boldsymbol{X}_{t-1:t+1}), \ldots, r(\boldsymbol{X}_{1:t+1})\} - 1)_+ q(x_{t+1} \mid \boldsymbol{X}_{1:t})}{\displaystyle\sum_{x_{t+1}}(\min\{r(x_{t+1}), r(\boldsymbol{X}_{t:t+1}), r(\boldsymbol{X}_{t-1:t+1}), \ldots, r(\boldsymbol{X}_{1:t+1})\} - 1)_+ q(x_{t+1} \mid \boldsymbol{X}_{1:t}) + 1 - p_t}
\end{aligned} \tag{A.92}
$$

Since $\min\{r(x_{t+1}), r(\boldsymbol{X}_{t:t+1}), r(\boldsymbol{X}_{t-1:t+1}), \ldots, r(\boldsymbol{X}_{1:t+1})\} \leq r(\boldsymbol{X}_{m(\boldsymbol{X}_{1:t+1})+1:t+1})$,

$$
\begin{aligned}
h_t^{\text{block}} &\leq \frac{\displaystyle\sum_{x_{t+1}}(r(\boldsymbol{X}_{m(\boldsymbol{X}_{1:t+1})+1:t+1}) - 1)_+ \, q(x_{t+1} \mid \boldsymbol{X}_{1:t})}{\displaystyle\sum_{x_{t+1}}(r(\boldsymbol{X}_{m(\boldsymbol{X}_{1:t+1})+1:t+1}) - 1)_+ \, q(x_{t+1} \mid \boldsymbol{X}_{1:t}) \;+\; 1 - p_t} \\[2em]
&\leq \frac{\displaystyle\sum_{x_{t+1}}(r(\boldsymbol{X}_{m(\boldsymbol{X}_{1:t+1})+1:t+1}) - 1)_+}{\displaystyle\sum_{x_{t+1}}(r(\boldsymbol{X}_{m(\boldsymbol{X}_{1:t+1})+1:t+1}) - 1)_+ \;+\; 1 - p_t}
\end{aligned} \tag{A.93}
$$

From equation A.83:

$$
\begin{aligned}
\sum_{x_{t+1}}(1 - r(\boldsymbol{X}_{m(\boldsymbol{X}_{1:t+1})+1:t+1})) &= q(\boldsymbol{X}_{m(t+1)+1:t}) - p(\boldsymbol{X}_{m(t+1)+1:t}) \\
&= (1 - r(\boldsymbol{X}_{m(t+1)+1:t})) q(\boldsymbol{X}_{m(t+1)+1:t}) \\
&\leq (1 - p_t) q(\boldsymbol{X}_{m(t+1)+1:t}) \\
&\leq (1 - p_t)
\end{aligned} \tag{A.94}
$$

Since

$$
\begin{aligned}
h_t^{\mathrm{branch}} &= \frac{\sum_{x_{t+1}}[r(\boldsymbol{X}_{m(\boldsymbol{X}_{1:t+1})+1:t+1}) - 1]_+}{\sum_{x_{t+1}}[r(\boldsymbol{X}_{m(\boldsymbol{X}_{1:t+1})+1:t+1}) - 1]_+ + \sum_{x_{t+1}}(1 - r(\boldsymbol{X}_{m(\boldsymbol{X}_{1:t+1})+1:t+1}))} \\
&\geq \frac{\sum_{x_{t+1}}[r(\boldsymbol{X}_{m(\boldsymbol{X}_{1:t+1})+1:t+1}) - 1]_+}{\sum_{x_{t+1}}[r(\boldsymbol{X}_{m(\boldsymbol{X}_{1:t+1})+1:t+1}) - 1]_+ + 1 - p_t} \\
&\geq h_t^{\mathrm{block}}
\end{aligned}
\tag{A.95}
$$

Table A.1: Comparison of different algorithm performance on GSM8K with Qwen-2.5. We list the average and standard deviation across 5 runs with different seeds.

| Method | Tokenwise | Blockwise | Ours |
|---|---|---|---|
| Block Efficiency | 6.40±0.10 | 6.51±0.09 | 6.64±0.04 |
| Decoding Speed | 31.52±0.06 | 31.70±0.05 | 32.61±0.02 |

Table A.2: Comparison of task performance across model sizes and methods.

| Metric | Method | 72B | 32B | 14B |
|---|---|---|---|---|
| GSM8K (Accuracy) | Tokenwise | 0.8213 | 0.8213 | 0.8327 |
| | HSD | 0.8517 | 0.8479 | 0.8327 |

Table A.3: Ablation on capping mechanism.

| Dataset | Method | ACC | BE | DS |
|---|---|---|---|---|
| GSM8K | HSD | 84.40±1.75% | 6.76±0.05 | 33.63±0.53 |
| HumanEval | HSD | 80.61±0.69% | 5.60±0.06 | 29.15±0.43 |
| GSM8K | HSD + Capping | 84.96±0.93% | 6.63±0.06 | 32.73±0.55 |
| HumanEval | HSD + Capping | 82.47±1.15% | 5.45±0.08 | 27.54±0.48 |

## E  EXTENDED EXPERIMENTS

**Result Robustness** To prove the robustness of our experiments and guarantee fair comparison, we conduct additional experiments with different methods as shown in Table A.1. We observe that our method demonstrates stable performance and exceeds both tokenwise and blockwise methods on average.

**Verification of Task Performance** We compare our method with the token-wise approach on GSM8K. As shown in Table A.2, our method achieves equivalent (or better) accuracy among different model sizes, demonstrating the preserved distributional fidelity.

**Capped Prefix Ratio Ablation Study** We conducted ablations on the role of the capped prefix ratio in Algorithm 2 (HSD), which is essential to preserve distributional fidelity, as shown in Table A.3. Removing capping (i.e., directly using the uncapped ratio to compute divergences) yields a slight increase in efficiency, but at the cost of varying degrees of performance degradation (which may or may not be obvious depending on the task).

## F  PYTHON IMPLEMENTATION

We provide the Python implementation of our Hierarchical Speculative Decoding (HSD) algorithm in Listing 2, which builds upon the token-wise speculative decoding approach from Hugging Face Wolf et al. (2020) Transformers v4.46.3, shown in Listing 1 for comparison. Following Hugging Face, our implementation eliminates the use of an explicit for-loop by leveraging an equivalent masking mechanism: we perform parallel sampling across all positions to determine whether to accept or reject subsequences of varying lengths, and then select the longest accepted prefix as the final output.

**Listing 1** Tokenwise Speculative Decoding (SD) `SD.py`

```python
import torch

def SD(candidate_input_ids, candidate_logits, new_logits):
    """
    Args:
        candidate_input_ids (Tensor): Token IDs from the draft model. Shape: [batch_size,
        ↪ seq_len]
        candidate_logits (Tensor): Logits from the draft model. Shape: [batch_size, seq_len,
        ↪ vocab_size]
        new_logits (Tensor): Logits from the target model. Shape: [batch_size, seq_len,
        ↪ vocab_size]
    Returns:
        n_matches (int): Number of accepted tokens from the draft model.
        valid_tokens (Tensor): Accepted token prefix with one new token sampled. Shape: [
        ↪ batch_size, n_matches+1]
    """

    # Convert logits to probabilities
    q = candidate_logits.softmax(dim=-1)
    p = new_logits.softmax(dim=-1)

    candidate_length = candidate_logits.shape[1]
    new_candidate_input_ids = candidate_input_ids[:, -candidate_length:]

    # Extract token-wise probabilities for the candidate tokens
    q_i = q[:, torch.arange(candidate_length), new_candidate_input_ids].squeeze(1)
    p_i = p[:, torch.arange(candidate_length), new_candidate_input_ids].squeeze(1)

    probability_ratio = p_i / q_i
    is_accepted = torch.rand_like(probability_ratio) <= probability_ratio

    # assuming batch size = 1
    n_matches = ((~is_accepted).cumsum(dim=-1) < 1).sum()  # this is `n` in algorithm 1

    # Next token selection: if there is a rejection, adopt the resampling distribution.
    if n_matches < candidate_length:
        p_n_plus_1 = p[:, n_matches, :]
        q_n_plus_1 = q[:, n_matches, :]
        p_prime = torch.clamp((p_n_plus_1 - q_n_plus_1), min=0)
        p_prime.div_(p_prime.sum())
    else:
        p_prime = p[:, n_matches, :]

    # Ensure we don't generate beyond max_len or an EOS token.
    if is_done_candidate[0] and n_matches == candidate_length:

        # Output length is assumed to be `n_matches + 1`. Since we won't generate another
        ↪ token with the target model
        # due to acceptance on EOS we fix `n_matches`
        n_matches -= 1
        valid_tokens = candidate_input_ids[:, -candidate_length:]

    else:
        # Next token selection: if there is a rejection, adjust the distribution from the main
        ↪  model before sampling.
        # The selected tokens include the matches (if any) plus the next sampled tokens
        if n_matches > 0:
            if n_matches < candidate_length:
                valid_tokens = candidate_input_ids[:, -candidate_length:n_matches -
        ↪ candidate_length]
                if not stop(valid_tokens, scores=None):
                    t = torch.multinomial(p_prime, num_samples=1)
                    valid_tokens = torch.cat(
                        (valid_tokens, t), dim=-1)
                else:
                    n_matches = n_matches-1
            else:
                valid_tokens = candidate_input_ids[:, -candidate_length:]
                if not stop(valid_tokens, scores=None):
                    t = torch.multinomial(p_prime, num_samples=1)
                    valid_tokens = torch.cat(
                    (valid_tokens, t), dim=-1)
                else:
                    n_matches = n_matches -1
        else:
            t = torch.multinomial(p_prime, num_samples=1)
            valid_tokens = t

    return valid_tokens, n_matches
```

**Listing 2** Hierarchical Speculative Decoding (HSD) `HSD.py`

```python
import torch

def HSD(candidate_input_ids, candidate_logits, new_logits):
    """
    Args:
        candidate_input_ids (Tensor): Token IDs from the draft model. Shape: [batch_size,
        ↪ seq_len]
        candidate_logits (Tensor): Logits from the draft model. Shape: [batch_size, seq_len,
        ↪ vocab_size]
        new_logits (Tensor): Logits from the target model. Shape: [batch_size, seq_len,
        ↪ vocab_size]
    Returns:
        n_matches (int): Number of accepted tokens from the draft model.
        valid_tokens (Tensor): Accepted token prefix with one new token sampled. Shape: [
        ↪ batch_size, n_matches+1]
    """

    # Convert logits to probabilities
    q = candidate_logits.softmax(dim=-1)
    p = new_logits.softmax(dim=-1)
    candidate_length = candidate_logits.shape[1]
    new_candidate_input_ids = candidate_input_ids[:, -candidate_length:]

    # Extract token-wise probabilities for the candidate tokens
    q_i = q[:, torch.arange(candidate_length), new_candidate_input_ids].squeeze(1)
    p_i = p[:, torch.arange(candidate_length), new_candidate_input_ids].squeeze(1)

    # Compute cumulative joint probabilities for draft and target model
    q_prev = torch.roll(q_i, shifts=1, dims=1)
    q_prev[:, 0] = 1.0
    q_cumprod = torch.exp(torch.log(q_prev).cumsum(dim=1)).unsqueeze(-1)
    q_next = q_cumprod * q[:, :candidate_length]
    p_prev = torch.roll(p_i, shifts=1, dims=1)
    p_prev[:, 0] = 1.0
    p_cumprod = torch.exp(torch.log(p_prev).cumsum(dim=1)).unsqueeze(-1)

    # Constrain p_cumprod with q_cumprod for computing the capped resampling distribution
    ratio = p_cumprod / q_cumprod
    previous_max = 1
    new_p_previous = torch.ones_like(p_cumprod).to(p_cumprod.device)
    for k in range(candidate_length):
        if ratio[:, k] > previous_max:
            previous_max = ratio[:, k]
        new_p_previous[:, k] = p_cumprod[:, k] / previous_max
    p_next = new_p_previous * p[:, :candidate_length]

    # Construct resampling distribution p'
    diffs = p_next - q_next
    p_plus = torch.clamp(diffs, min=0.0)
    p_minus = torch.clamp(-diffs, min=0.0)
    p_primes = p_plus / torch.maximum(p_plus.sum(dim=-1, keepdim=True), p_minus.sum(dim=-1,
     ↪ keepdim=True))

    # Step-back probability: reject prefix with 1 - mass of p'
    step_back_probs = 1 - p_primes.sum(dim=-1)
    step_back = torch.rand_like(step_back_probs) < step_back_probs

    # Find first position to stop (from the end)
    if step_back.all():
        stop_positions = 0
    else:
        stop_positions = candidate_length - n_matches - 1 - torch.flip(~step_back, [-1]).max
     ↪ (-1, keepdim=True)[1]

    # Mask to decide which tokens are accepted
    select = torch.zeros_like(step_back).to(step_back.device)

    # apply cumprod on the ratio instead of the raw probabilities to avoid underflow
    probability_ratio = (p_i / q_i).cumprod(1).unsqueeze(-1)
    is_accepted = torch.rand_like(probability_ratio) <= probability_ratio

    # only decide to accept or not at the last position based on the joint probability ratio
    # assign 0 to all positions when the full draft is rejected, otherwise assign 1 to the
     ↪ rest of the positions
    select[torch.arange(p_primes.shape[0]), stop_positions] = ~is_accepted[:, -1:]
    is_accepted = 1 - torch.cumsum(select, dim=-1)

    #### assume batch_size=1 for the current implementation
    n_matches = is_accepted.sum().item()
```

**Listing 2** Hierarchical Speculative Decoding `HSD.py` (cont.)

```
1        if is_done_candidate[:] and n_matches == candidate_length:
2            # Output length is assumed to be 'n_matches + 1'. Since we won't generate another
          ↪ token with the target model
3            # due to acceptance on EOS we fix 'n_matches'
4            n_matches -= 1
5            # valid_tokens = new_candidate_input_ids[:, : n_matches + 1]
6            valid_tokens = candidate_input_ids[:, -candidate_length:]
7
8        else:
9            # Next token selection: if there is a rejection, adjust the distribution from the main
          ↪  model before sampling.
10           gamma = candidate_length
11           p_n_plus_1 = p[:, candidate_length, :]
12           if n_matches < gamma:
13               p_prime = p_primes[:, n_matches]
14               p_prime = p_prime/p_prime.sum(-1, keepdim=True)
15           else:
16               p_prime = p_n_plus_1
17
18           # The selected tokens include the matches (if any) plus the next sampled tokens
19           # because if n_matches=0, we add one resampled token for sure, if n_matches=10, we add
          ↪  one more for sure
20           # as well, because the previous if checked not stop and n_matches-candidate_length
          ↪ will be 0 causing problem
21           if n_matches > 0 and n_matches<candidate_length:
22               valid_tokens = candidate_input_ids[:, -candidate_length:n_matches-candidate_length
                ↪ ]
23               if not stop(candidate_input_ids[:, :n_matches-candidate_length], scores=None):
24                   t = torch.multinomial(p_prime, num_samples=1)
25                   valid_tokens = torch.cat(
26                       (valid_tokens, t), dim=-1)
27               else:
28                   n_matches = n_matches-1
29           else:
30               t = torch.multinomial(p_prime, num_samples=1)
31               if n_matches==0:
32                   valid_tokens = t
33               else:
34                   valid_tokens = candidate_input_ids[:, -candidate_length:]
35                   valid_tokens = torch.cat(
36                       (valid_tokens, t), dim=-1)
37
38       return valid_tokens, n_matches
```

## G    INTEGRATION WITH RECURSIVE REJECT SAMPLING IN THE MULTI-DRAFT SETUP

We demonstrate in Algorithm 3 that our HSD algorithm is compatible with existing lossless multi-draft verification methods, exemplified by Recursive Reject Sampling (RRS) with replacement Yang et al. (2024). Notably, independently sampled parallel draft sequences do not guarantee the existence of an additional draft sequence that shares the accepted subsequence as its prefix.

---

**Algorithm 3** Hierarchical Speculative Sampling with Recursive Rejection Sampling

---

**Require:** Draft tokens: $\boldsymbol{X}_{1:t}^k = \{x_1^k, ..., x_\gamma^k\}_{k=1}^K$;

   Target probabilities for all draft tokens: $\{p(\cdot), ..., p(\cdot|\boldsymbol{X}_{1:\gamma}^k)\}_{k=1}^K$;

   Draft probabilities for all draft tokens: $\{q(\cdot), ..., q(\cdot|\boldsymbol{X}_{1:\gamma}^k)\}_{k=1}^K$;

1: Initialize $\tau = 0$;
2: Initialize $\{x_i^1\}_1^\gamma$;
3: **for** $k$ **in** $1 : K$ **do**
4:    **if** $\boldsymbol{X}_{1:\tau} = \boldsymbol{X}_{1:\tau}^k$ **then**
5:      **for** $j$ **in** $\tau + 1 : \gamma$ **do**
6:        Set $x_j = x_j^k$                          *#select draft $\boldsymbol{X}_{\tau+1:\gamma}^k$ for verification*
7:      **end for**
8:
9:      **for** $t$ **in** $\gamma : \tau + 1$ **do**
10:        Compute acceptance probability $h_t$ from Equation (19) based on the corresponding probabilities for the draft tokens: $\{x_{\tau+1}, ..., x_\gamma\}$
11:        Sample $\eta_t \sim U(0,1)$
12:        **if** $h_t \geq \eta_t$ **then**
13:          Set $\tau = t$
14:          **break**
15:        **else**
16:          Set $\tau = t - 1$
17:          **continue**
18:        **end if**
19:      **end for**
20:    **else**
21:      **continue**                       *#skip draft $\boldsymbol{X}_{1:\gamma}^k$ due to prefix mismatch*
22:    **end if**
23:
24:    **if** $\tau = \gamma$ **then**
25:      Sample token from $p(\cdot|\boldsymbol{X}_{1:\gamma})$   *#accept the entire selected draft and sample a bonus token*
26:      **break**
27:    **else**
28:      Compute $P_{\text{res}}^*(\cdot \mid \boldsymbol{X}_{1:\tau})$;
29:      Set $p(\cdot|\boldsymbol{X}_{1:\tau}) = P_{\text{res}}^*(\cdot \mid \boldsymbol{X}_{1:\tau})$;       *#set $P_{res}^*(\cdot \mid \boldsymbol{X}_{1:\tau})$ as new target distribution*
       Set $r(\cdot|\boldsymbol{X}_{1:\tau}) = \frac{P_{\text{res}}^*(\cdot|\boldsymbol{X}_{1:\tau})}{q(\tilde{x}|\boldsymbol{X}_{1:\tau})}$         *#set $r(\cdot \mid \boldsymbol{X}_{1:\tau})$ as new probability ratio*
30:    **end if**
31:
32: **end for**
   Sample token from $P_{\text{res}}^*(\cdot \mid \boldsymbol{X}_{1:\tau})$
**Ensure:** $[\boldsymbol{X}_{1:\tau}, \text{token}]$

---

## H    COMPUTATION EFFICIENCY

We begin by noting that the computational cost of the verification stage in HSD is *effectively as efficient as that of tokenwise verification* in practice.

While there are minor differences in the computational cost of verification—whether using any of the three verification methods—these differences are **insignificant in practice** compared to the reduction in target model forward passes. Indeed, **block efficiency (or equivalently, the acceptance rate) remains the most meaningful metric for evaluating performance**. A detailed complexity analysis is provided in the revised version below:

Both HSD (Eqs. 17–20 in our paper) and blockwise verification (Eqs. 4–5 in Sun et al. (2024)) require summing over the vocabulary to compute the acceptance probability at each position. Therefore, HSD introduces **no theoretical overhead** compared to blockwise verification.

Moreover, our implementation is **more efficient** than that of blockwise verification (Appendix A in Sun et al. (2024) ). By leveraging an equivalent masking mechanism, we eliminate the for-loop and compute probabilities for all positions in parallel. This makes HSD nearly as efficient as tokenwise verification. While tokenwise verification only sums over the vocabulary at a rejected position, **our HSD computation is fully parallelized via tensor operations across both the vocabulary and the draft length $\gamma$, which is typically much smaller than the vocabulary size $\mathcal{V}$.**

Importantly, the computational cost of verification is negligible relative to the reduction in target model forward passes, which is the main bottleneck in verification. Below, we compare the verification cost with the forward-pass reduction for a batch size of 1.

Since the vocabulary size $\mathcal{V}$ is much larger than the draft length $\gamma$, the main cost of HSD arises from computing branch divergences, which requires only $4\gamma\mathcal{V}$ FLOPs:

1. $r^*(\mathbf{X}_{1:t}) - 1 \to \gamma\mathcal{V}$ FLOPs
2. Selecting $(r^*(\mathbf{X}_{1:t}) - 1 > 0)$ via the $\max$ operator $\to \gamma\mathcal{V}$ FLOPs
3. Multiplication by $q \to \gamma\mathcal{V}$ FLOPs
4. Summation over the vocabulary $\to \gamma\mathcal{V}$ FLOPs

For Qwen-2.5 with $|\mathcal{V}| = 151{,}643$ and $\gamma = 10$, this amounts to approximately **5.8M FLOPs**.

In contrast, **the forward-pass FLOPs per new token** in large language models are orders of magnitude larger, even with KV-cache inference. Considering the main contributions:

- $Q \cdot K$ dot products
- Attention score $\times V$
- All projection/MLP FLOPs
- Ignoring softmax, LayerNorm, rotary embeddings, and bias

The per-token FLOPs can be approximated as:

$$\text{FLOPs per new token} \approx L \cdot \left[4dL_{\text{past}} + 4d^2 + 2dd_{\text{ff}}\right],$$

where $L$ is the number of transformer layers, $d$ is the hidden size, $d_{\text{ff}}$ is the MLP intermediate size, and $L_{\text{past}}$ is the number of cached tokens.

Using a context length $L_{\text{past}} = 1024$, the per-token FLOPs for Qwen2.5 models are:

- Qwen2.5-0.5B: **0.374 GFLOPs**
- Qwen2.5-72B: **62.915 GFLOPs**

As shown in Table 1 in our paper, all methods achieve block efficiency larger than 2. Consequently, the cost of HSD verification is negligible relative to the reduction in target model forward passes.

To directly quantify the overhead of the verification stage, we evaluate verification cost on 100 GSM8K problems using GPTQ-quantized 8-bit Qwen2.5-72B-Instruct and Qwen2.5-0.5B-Instruct

as target and draft models on a single H200 GPU. As shown in Table A.4, the verification stage consistently accounts for **less than 1%** of the total decoding time. At the same time, the vast majority of runtime is spent in the draft and target forward passes. Here, the draft forward pass accounts for about 24% of the runtime, and the target forward pass accounts for about 72% in both blockwise and HSD. The verification stage of HSD is about 20% faster than that of blockwise.

Table A.4: Runtime breakdown of Blockwise and HSD.

| Component | Blockwise Mean (ms/token) | Blockwise % | HSD Mean (ms/token) | HSD % |
|---|---|---|---|---|
| **Total** | **34.168** | **100.00%** | **33.788** | **100.00%** |
| Prefill | 0.913 | 2.67% | 0.915 | 2.71% |
| Draft Forward | 8.210 | 24.03% | 7.865 | 23.28% |
| Target Forward | 24.690 | 72.26% | 24.695 | 73.09% |
| KV Cache Input | 0.007 | 0.02% | 0.005 | 0.01% |
| KV Cache Output | 0.070 | 0.20% | 0.067 | 0.20% |
| Logits Processing | 0.093 | 0.27% | 0.103 | 0.31% |
| Verification | 0.160 | 0.47% | 0.127 | 0.37% |
| Other | 0.025 | 0.08% | 0.012 | 0.03% |

