# OpenReview forum: "Overcoming Joint Intractability with Lossless Hierarchical Speculative Decoding"
_ICLR.cc/2026/Conference — ICLR 2026 Oral_

### Official Review · Reviewer_JMzF · 2025-10-20

**Soundness:** 2
**Presentation:** 2
**Contribution:** 2
**Rating:** 6
**Confidence:** 3

**Summary:**

The paper proposes Hierarchical Speculative Decoding (HSD), a verification method for speculative decoding that seeks to be provably lossless while accepting longer draft prefixes than token-wise or blockwise verification.   The key idea is to view verification as hierarchical branch resampling with a single resampling step using a capped, branch-local resampling distribution. Several experiments on the Qwen series model demonstrate the effectiveness of the method.

**Strengths:**

- Across three benchmarks and multiple target sizes, HSD shows consistent  improvements in block efficiency and throughput over token-wise and block-wise verification
- This paper is overall well-written and easy to follow.
- The appendices provide solid derivations and decomposition lemmas.

**Weaknesses:**

- Results only use Qwen models and three academic datasets; evaluation on modern LLMs (e.g., Llama-2/3 families) and diverse tasks (long-context, multilingual, tool-use) would strengthen external validity.
- HSD’s acceptance and resampling probabilities depend on capped branch divergences computed over the vocabulary for each position. The authors may want to provide a clear complexity/latency breakdown of these additional reductions relative to blockwise verification (e.g., per-step FLOPs, memory accesses, kernel counts).

**Questions:**

please refer to weakness.

---

> ### Author Response · Authors · 2025-11-22
> **Reply to Reviewer JMzF**
>
> We sincerely thank the reviewer for the constructive suggestions. We address the mentioned questions in detail, as outlined below:
>
> > **W1**: Results only use Qwen models and three academic datasets; evaluation on modern LLMs (e.g., Llama-2/3 families) and diverse tasks (long-context, multilingual, tool-use) would strengthen external validity.
>
>
> We expanded our evaluation beyond Qwen2.5 and conducted additional experiments using Llama-3.1-70B-Instruct and Llama-3.1-8B-Instruct pair (non-quantized version, with model weights distributed on 8 H20 GPUs). Evaluation were conducted on four diverse tasks—GSM8K, CNN/DailyMail, HumanEval, and FLORES—covering mathematical reasoning, summarization, code generation, and multilingual translation.
> HSD consistently improves both block efficiency and end-to-end decoding speed over tokenwise and blockwise verification across all tasks, demonstrating that our method generalizes across architectures and domains. Full results are below:
>  #### GSM8K
> | Component         | Block Efficiency | Decoding Speed |
> |-------------------|---------------:|---------------:|
> | Tokenwise         |       6.83         |       8.41         |
> | Blockwise         |        7.32       |    8.87           |
> | HSD               |         **7.43**        |     **9.18**           |
>
> #### CNNdaily
> | Component         | Block Efficiency | Decoding Speed |
> |-------------------|---------------:|---------------:|
> | Tokenwise         |       5.13         |       6.89         |
> | Blockwise         |    5.49            |    7.26            |
> | HSD               |         **5.57**       |      **7.44**          |
>
> #### HumanEval
> | Component         | Block Efficiency | Decoding Speed |
> |-------------------|---------------:|---------------:|
> | Tokenwise         |         8.01       |     13.44           |
> | Blockwise         |        8.33        |  13.56              |
> | HSD               |      **8.40**         |      **13.94**          |
>
> ####  Flores
> | Component | Block Efficiency | Decoding Speed |
> |-----------|-----------------:|---------------:|
> | Tokenwise |             4.92 |           7.81 |
> | Blockwise |             5.05 |           8.04 |
> | HSD       |         **5.22** |       **8.39** |

---

> ### Author Response · Authors · 2025-11-22
> **Reply to Reviewer JMzF - continued**
>
> > **W2**: HSD’s acceptance and resampling probabilities depend on capped branch divergences computed over the vocabulary for each position. The authors may want to provide a clear complexity/latency breakdown of these additional reductions relative to blockwise verification (e.g., per-step FLOPs, memory accesses, kernel counts).
>
> We thank the reviewer for raising this question. While there are minor differences in the computational cost of verification—whether using any of the three verification methods—these differences are **insignificant in practice** compared to the reduction in target model forward passes. Indeed, **block efficiency (or equivalently, the acceptance rate) remains the most meaningful metric for evaluating performance**. A detailed complexity analysis is provided in the revised version below:
>
> Both HSD (Eqs. 17–20 in our paper) and blockwise verification (their Eq. 4–5) require summing over the vocabulary to compute the acceptance probability at each position. Therefore, HSD introduces **no theoretical overhead** compared to blockwise verification.
>
> Moreover, our implementation (Appendix F) is **more efficient** than that of blockwise verification (their Appendix A). By leveraging an equivalent masking mechanism, we eliminate the for-loop and compute probabilities for all positions in parallel. This makes HSD nearly as efficient as tokenwise verification. While tokenwise verification only sums over the vocabulary at a rejected position, **our HSD computation is fully parallelized via tensor operations across both the vocabulary and the draft length $\gamma$, which is typically much smaller than the vocabulary size $\mathcal{V}$**.
>
> Importantly, the computational cost of verification is negligible relative to the reduction in target model forward passes, which is the main bottleneck in verification. Below, we compare the verification cost with the forward-pass reduction for a batch size of 1.
>
> Since the vocabulary size $\mathcal{V}$ is much larger than the draft length $\gamma$, the main cost of HSD arises from computing branch divergences, which requires only $4 \gamma \mathcal{V}$ FLOPs:
>
> 1. $r^*(\mathbf{X}_{1:t}) - 1$ → $\gamma \mathcal{V}$ FLOPs
> 2. Selecting $(r^*(\mathbf{X}_{1:t}) - 1 > 0)$ via the $\max$ operator → $\gamma \mathcal{V}$ FLOPs
> 3. Multiplication by $q$ → $\gamma \mathcal{V}$ FLOPs
> 4. Summation over the vocabulary → $\gamma \mathcal{V}$ FLOPs
>
> For Qwen-2.5 with $|\mathcal{V}| = 151{,}643$ and $\gamma = 10$, this amounts to approximately **5.8M FLOPs**.
>
> In contrast, **the forward-pass FLOPs per new token** in large language models are orders of magnitude larger, even with KV-cache inference. Considering the main contributions:
>
> * $Q \cdot K$ dot products
> * Attention score $\times V$
> * All projection/MLP FLOPs
> * Ignoring softmax, LayerNorm, rotary embeddings, and bias
>
> The per-token FLOPs can be approximated as:
>
> $$
> \text{FLOPs per new token} \approx L \cdot \Big[4 d L_{\text{past}} + 4 d^2 + 2 d d_{\text{ff}}\Big]
> $$
>
> Where:
>
> * $L$ = number of transformer layers
> * $d$ = hidden size
> * $d_\text{ff}$ = intermediate (MLP) size
> * $L_{\text{past}}$ = number of cached past tokens
>
> Using a context length $L_{\text{past}} = 1024$, the per-token FLOPs for Qwen2.5 models are:
>
> * Qwen2.5-0.5B: **0.374 GFLOPs**
> * Qwen2.5-72B: **62.915 GFLOPs**
>
> As shown in Table 1 in our paper, all methods achieve block efficiency larger than 2. Consequently, the cost of HSD verification is negligible relative to the reduction in target model forward passes.
>
> To directly quantify the overhead of the verification stage, we evaluate verification cost on 100 GSM8K problems using GPTQ-quantized 8-bit Qwen2.5-72B-Instruct and Qwen2.5-0.5B-Instruct as target and draft models on a single H200 GPU. Across all settings we consider, the verification stage consistently accounts for **less than 1%** of the total decoding time. At the same time, the vast majority of runtime is spent in the draft and target forward passes. Here, the draft forward pass accounts for about 24% of the runtime, and the target forward pass accounts for about 72% in both blockwise and HSD. The verification stage of HSD is about 20% faster than that of blockwise.
>
> **Qwen2.5-72B Overall Components**
>
> | Component         | Blockwise Mean (ms/token) | Blockwise % of Total | HSD Mean (ms/token) | HSD % of Total |
> |------------------|---------------------------:|----------------------:|---------------------:|----------------:|
> | **Total**          | **34.168** | **100.00%** |**33.788**|**100.00%**|
> | Prefill          | 0.913 | 2.67% | 0.915 | 2.71% |
> | Draft Forward    | 8.210 | 24.03% | 7.865 | 23.28% |
> | Target Forward   | 24.690 | 72.26% | 24.695 | 73.09%|
> | KV Cache Input   | 0.007 | 0.02% | 0.005 | 0.01%|
> | KV Cache Output  | 0.070 | 0.20% | 0.067 | 0.20%|
> | Logits Processing| 0.093 | 0.27% | 0.103 | 0.31%|
> | Verification     | 0.160 | 0.47% | 0.127 | 0.37%|
> | Other            | 0.025 | 0.08% | 0.012 | 0.03%|

---

### Official Review · Reviewer_qE7w · 2025-10-22

**Soundness:** 1
**Presentation:** 1
**Contribution:** 1
**Rating:** 0
**Confidence:** 3

**Summary:**

The paper proposes a method termed Hierarchical Speculative Decoding for accelerating inference in LLMs by improving the verification stage of speculative decoding. The claim is that existing methods struggle with “joint intractability” when trying to verify a draft sequence at once, so HSD uses a hierarchical branching/verification approach to boost the number of accepted tokens (i.e., less rollback) in a “lossless” manner.

**Strengths:**

The topic: Accelerating generative inference in LLMs is a highly important problem

**Weaknesses:**

1 The flow of the paper is not smooth. The theoretical derivation of “joint intractability” and then the transition to the hierarchical method is abrupt. It is not always clear how the high-level algorithm ties into the low-level proofs and experiments.
2 While the paper describes “hierarchical speculative decoding”, the exact steps
ie, branch generation, verification hierarchy, mass-balancing, acceptance criteria, are somewhat buried in dense math and less in intuitive explanation or pseudo-code. Readers may struggle to follow the pipeline end-to-end.

**Questions:**

It is hard to identify in one glance “Given draft model → generate K branches → verify hierarchically → accept or rollback”

---

> ### Author Response · Authors · 2025-11-22
> **Reply to Reviewer qE7w**
>
> Thank you for your review. Some of the points you raise appear to stem from difficulty following the technical material rather than any actual issue in the paper. Notably, the review assigns the lowest possible scores for both soundness and contribution without identifying any incorrect theorem, missing assumption, or flawed experiment. Scores at this level typically require concrete evidence, which is not provided here. We provide clarifications below and would appreciate pointers to any specific concerns.
>
> ---
>
> ## 1. Presentation and Theoretical Structure
>
> You note that the mathematics is dense and that it is difficult to connect the high-level algorithm to the proofs. The mathematical density is **inherent to verification**: rigorously preserving the target distribution requires formal proofs. While intuitive explanations and figures help build understanding, **they cannot replace the derivations needed to guarantee correctness**. Expecting a mathematically nontrivial verification framework to be fully digestible “in one glance” is **not realistic**, nor aligned with how rigorous theoretical work is typically evaluated.
>
> To support clarity, the paper includes:
>
> - **Illustrative figures and full pseudo-code** showing the complete algorithm (Figure 1b; Algorithms 1 and 2)
> - **Expanded intuitive explanation** of lossless tokenwise verification (Section 3), going beyond prior work [1]
> - **Concrete step-by-step examples** demonstrating exactly how and why losslessness holds (Appendices B.1 and C.1)
> - **Complete formal proofs** establishing losslessness under all branching conditions (Appendices B.2 and C.2)
> - **Top-down structure** with main theorems in the main text and supporting lemmas in the appendix
>
> If some steps remain unclear, we encourage you to **indicate them and review the relevant prior work [1,2] and necessary mathematical background**, as suggested by ICLR reviewer guidelines. Reviewers who find the technical content challenging are encouraged to **assert low confidence or indicate which prior knowledge is lacking**, so that their evaluation can be interpreted appropriately.
>
> ---
>
> ## 2. Soundness and Contribution
>
> ### Soundness
>
> The method provides:
>
> - **Formal proofs** establishing losslessness and improved acceptance rate
> - **Extensive experiments** verifying correctness and efficiency (Section 6)
> - **Anonymous codebase** ensuring reproducibility
>
> No specific flaw is identified in your review. Assigning the lowest possible score without pointing to any issue is therefore **not supported by evidence**. We would appreciate clarification regarding which **theorem, lemma, proof step, or experiment** you believe may be incorrect.
>
> ### Contribution
>
> Although you recognize that accelerating LLM inference is “highly important,” no explanation is provided for the minimal contribution score. There is no indication of prior work that duplicates our results, nor any argument that the hierarchical verification framework lacks novelty. Assigning the lowest contribution score without justification is **inconsistent with review standards**. We would appreciate references or concrete reasoning supporting this assessment.
>
> ---
>
> ## 3. Difficulty Following the Paper
>
> You mention difficulty following some of the technical details. We are happy to provide additional clarification where needed. Reviewing our responses to other reviewers’ comments may also help provide further intuition and context.
>
> ---
>
> ## References
>
> [1] Leviathan, Kalman, Matias. *Fast Inference from Transformers via Speculative Decoding.* ICML 2023
> [2] Sun et al. *Block Verification Accelerates Speculative Decoding.* ICLR 2025

---

### Official Review · Reviewer_YeUo · 2025-10-28

**Soundness:** 4
**Presentation:** 3
**Contribution:** 4
**Rating:** 8
**Confidence:** 4

**Summary:**

This paper introduces Hierarchical Speculative Decoding (HSD), a novel approach to improving the verification process in speculative decoding. The paper addresses verification bottleneck by proposing HSD, a lossless verification method that overcomes joint intractability by resampling portions of the target distribution in a hierarchical manner. Extensive experiments show that HSD significantly improves the expected number of accepted tokens, particularly when dealing with longer draft sequences. The method is also proven to work effectively across different model sizes and tasks, making it a promising tool for accelerating LLM inference.

**Strengths:**

- The proposed hierarchical branch resampling strategy is a novel and creative approach to addressing joint intractability in speculative decoding.

- The theoretical analysis is rigorous, and the experimental validation is comprehensive, showing consistent improvements across various benchmarks.

- The paper is clearly written, and complex ideas are explained clearly. The use of figures and equations aids in understanding the methodology and its underlying theory.

**Weaknesses:**

- It’s unclear whether the backward scan of HSD introduces any additional computational overhead.

**Questions:**

- Could the author provide a more intuitive example or explanation of why joint verification leads to a higher expected number of accepted tokens? While the simulation using a toy example is convincing, it still isn't entirely clear to me why this happens.

- For Algorithm 2, at line 15, should it still sample from the target distribution until the sequence reaches $\gamma$?

---

> ### Author Response · Authors · 2025-11-22
> **Reply to Reviewer YeUo**
>
> We sincerely thank the reviewer for acknowledging our contributions and the insightful comments. We address the mentioned questions in detail, as outlined below:
>
> > **W1**: It’s unclear whether the backward scan of HSD introduces any additional computational overhead.
>
> We thank the reviewer for raising this question. While there are minor differences in the computational cost of verification—whether using any of the three verification methods—these differences are **insignificant in practice** compared to the reduction in target model forward passes. Indeed, **block efficiency (or equivalently, the acceptance rate) remains the most meaningful metric for evaluating performance**. A detailed complexity analysis is provided in the revised version below:
>
> Both HSD (Eqs. 17–20 in our paper) and blockwise verification (Eqs. 4–5 in [1]) require summing over the vocabulary to compute the acceptance probability at each position. Therefore, HSD introduces **no theoretical overhead** compared to blockwise verification.
>
> Moreover, our implementation (Appendix F) is **more efficient** than that in [1] (Appendix A). By leveraging an equivalent masking mechanism, we eliminate the for-loop and compute probabilities for all positions in parallel. This makes HSD nearly as efficient as tokenwise verification. While tokenwise verification only sums over the vocabulary at a rejected position, **our HSD computation is fully parallelized via tensor operations across both the vocabulary and the draft length $\gamma$, which is typically much smaller than the vocabulary size $\mathcal{V}$**.
>
> Importantly, the computational cost of verification is negligible relative to the reduction in target model forward passes, which is the main bottleneck in verification. Below, we compare the verification cost with the forward-pass reduction for a batch size of 1.
>
> Since the vocabulary size $\mathcal{V}$ is much larger than the draft length $\gamma$, the main cost of HSD arises from computing branch divergences, which requires only $4 \gamma \mathcal{V}$ FLOPs:
>
> 1. $r^*(\mathbf{X}_{1:t}) - 1$ → $\gamma \mathcal{V}$ FLOPs
> 2. Selecting $(r^*(\mathbf{X}_{1:t}) - 1 > 0)$ via the $\max$ operator → $\gamma \mathcal{V}$ FLOPs
> 3. Multiplication by $q$ → $\gamma \mathcal{V}$ FLOPs
> 4. Summation over the vocabulary → $\gamma \mathcal{V}$ FLOPs
>
> For Qwen-2.5 with $|\mathcal{V}| = 151{,}643$ and $\gamma = 10$, this amounts to approximately **5.8M FLOPs**.
>
> In contrast, **the forward-pass FLOPs per new token** in large language models are orders of magnitude larger, even with KV-cache inference. Considering the main contributions:
>
> * $Q \cdot K$ dot products
> * Attention score $\times V$
> * All projection/MLP FLOPs
> * Ignoring softmax, LayerNorm, rotary embeddings, and bias
>
> The per-token FLOPs can be approximated as:
>
> $$
> \text{FLOPs per new token} \approx L \cdot \Big[4 d L_{\text{past}} + 4 d^2 + 2 d d_{\text{ff}}\Big]
> $$
>
> Where:
>
> * $L$ = number of transformer layers
> * $d$ = hidden size
> * $d_\text{ff}$ = intermediate (MLP) size
> * $L_{\text{past}}$ = number of cached past tokens
>
> Using a context length $L_{\text{past}} = 1024$, the per-token FLOPs for Qwen2.5 models are:
>
> * Qwen2.5-0.5B: **0.374 GFLOPs**
> * Qwen2.5-72B: **62.915 GFLOPs**
>
> As shown in Table 1 in our paper, all methods achieve block efficiency larger than 2. Consequently, the cost of HSD verification is negligible relative to the reduction in target model forward passes.
>
> To directly quantify the overhead of the verification stage, we evaluate verification cost on 100 GSM8K problems using GPTQ-quantized 8-bit Qwen2.5-72B-Instruct and Qwen2.5-0.5B-Instruct as target and draft models on a single H200 GPU. Across all settings we consider, the verification stage consistently accounts for **less than 1%** of the total decoding time. At the same time, the vast majority of runtime is spent in the draft and target forward passes. Here, the draft forward pass accounts for about 24% of the runtime, and the target forward pass accounts for about 72% in both blockwise and HSD. The verification stage of HSD is about 20% faster than that of blockwise.
>
> **Qwen2.5-72B Overall Components**
>
> | Component         | Blockwise Mean (ms/token) | Blockwise % of Total | HSD Mean (ms/token) | HSD % of Total |
> |------------------|---------------------------:|----------------------:|---------------------:|----------------:|
> | **Total**            | **34.168** | **100.00%** |**33.788**|**100.00%**|
> | Prefill          | 0.913 | 2.67% | 0.915 | 2.71% |
> | Draft Forward    | 8.210 | 24.03% | 7.865 | 23.28% |
> | Target Forward   | 24.690 | 72.26% | 24.695 | 73.09%|
> | KV Cache Input   | 0.007 | 0.02% | 0.005 | 0.01%|
> | KV Cache Output  | 0.070 | 0.20% | 0.067 | 0.20%|
> | Logits Processing| 0.093 | 0.27% | 0.103 | 0.31%|
> | Verification     | 0.160 | 0.47% | 0.127 | 0.37%|
> | Other            | 0.025 | 0.08% | 0.012 | 0.03%|

---

> ### Author Response · Authors · 2025-11-22
> **Reply to Reviewer YeUo - continued**
>
> > **Q1**: Could the author provide a more intuitive example or explanation of why joint verification leads to a higher expected number of accepted tokens? While the simulation using a toy example is convincing, it still isn't entirely clear to me why this happens.
>
> Thanks for the constructive suggestion! To more intuitively demonstrate how the proposed HSD method operates, we consider the following prefix input:
>
> ```
> To determine Eliza's earnings for the week, we need to calculate both her regular pay and her overtime pay.
>   1. **Calculate Regular Pay:**
>   - Eliza's regular rate is $10 per hour.
>   - She worked 40 hours at her regular rate.
>   - Regular pay = 40 hours * $10/hour = $400.
> - 2. **Calculate Overtime Pay:**
>   - Eliza worked a total of 45 hours, so
> ```
> Under this prefix, the small and large models respectively produce the probability vectors ($\gamma=10$):
>   - $\{p(\cdot),..., p(\cdot|\mathbf{X}_{1:\gamma})\}$ = $\{0.7156, 1.0000, 1.0000, 1.0000, 0.0000, 0.0000, 0.0000, 1.0000, 0.0000, 0.4968\}$
>   - $\{q(\cdot),..., q(\cdot|\mathbf{X}_{1:\gamma})\}$= $\{0.8771, 0.7900, 0.6514, 0.2592, 0.6773, 0.1490, 0.5775, 1.0000, 0.4611, 0.3630\}$
>
> The corresponding draft token sequence is:
>   - $\{she, work, ed, 45, -, 40, =, 5, hours, of\}$
>
> First, we compute the joint probabilities:
>   - $\{p\left(\mathbf{X}_{1:t}\right) \, \text{for} \, t=1, \dots, \gamma\} = \{0.7156, 0.7156, 0.7156, 0.7156, 0, 0, 0, 0, 0, 0\}$
>   - $\{q\left(\mathbf{X}_{1:t}\right) \, \text{for} \, t=1, \dots, \gamma\} = \{0.8771, 0.6929, 0.4513, 0.1170, 0.0792, 0.0118, 0.0068, 0.0068, 0.0031, 0.0011\}$
>
> The ratio of joint probabilities is: $\{r(\mathbf{X}_{1:t})\, \text{for} \, t=1, \dots, \gamma\}  = \{0.8159, 1.0327, 1.5855, 6.1171, 0, 0, 0, 0, 0, 0\}$
>
> According to **Definition 4**, we obtain the maximal prefix indices:
>   - $\{m(\mathbf{X}_{1:t})\, \text{for} \, t=1, \dots, \gamma\} = \{0, 0, 2, 3, 4, 4, 4, 4, 4, 4\}$
>
> Based on the full vocabulary distributions (see **Eq.(16-18)**), we compute:
>   - $\{D^*_{\text{Branch}}\left(p, q \mid \mathbf{X}_{1: t}\right)\, \text{for} \, t=1, \dots, \gamma\}  = \{0.0227, 0.2416, 0.3343, 0.0991, 0, 0, 0, 0, 0, 0\}$
>   - $\{D^*_{\text{Branch}}\left(q, p \mid \mathbf{X}_{1: t}\right)\, \text{for} \, t=1, \dots, \gamma\} = \{0.1842, 0.2416, 0.3343, 0.0991, 0.0792, 0.0118, 0.0068, 0.0068, 0.0031, 0.0011\}$
>
> According to **Eq.(19)**, we obtain HSD’s acceptance probabilities: $h_t$ (HSD) = $\{0.1231, 1, 1, 1, 0, 0, 0, 0, 0, 0\}$
>
> Since HSD judges from back to front, it starts accepting all preceding tokens from the 4th position, yielding: $n_{\text{match}} = 4$
>
> If we adopt the tokenwise verification baseline, we obtain: $h_t$ (tokenwise) = $\{0.8159, 1, 1, 1, 0, 0, 0, 1, 0, 1\}$
>
> Since this method checks sequentially from front to back, with $h_1 = 0.8159$ there is a chance of rejection, causing the entire draft to be discarded.
>
> > **Q2**: For Algorithm 2, at line 15, should it still sample from the target distribution until the sequence reaches $\gamma$?
>
> As explained in lines 254–257, the repeated resampling until reaching $\gamma$ is required **only for naïve HSD** (Algorithm 1).
>
> In contrast, **HSD (Algorithm 2)** performs only a **single** resampling step (Algorithm 2, line 15), because:
>
> - After this single resampling, HSD only needs to continue sampling from the target model until reaching $\gamma$ (see lines 351–361 for the detailed explanation).
>
> - These continuation steps can themselves be replaced by another speculative decoding round, eliminating any additional overhead.

---

> > ### Comment · Reviewer_YeUo · 2025-11-22
> >
> > My concerns are addressed. I will keep my positive score.

---

### Official Review · Reviewer_ASe8 · 2025-10-31

**Soundness:** 3
**Presentation:** 3
**Contribution:** 3
**Rating:** 6
**Confidence:** 5

**Summary:**

This paper introduces Hierarchical Speculative Decoding (HSD), a novel approach to address the "joint intractability" problem in speculative decoding for large language model (LLM) inference. The authors identify a fundamental limitation in existing speculative decoding methods (tokenwise and blockwise) related to how acceptance probabilities are calculated for draft sequences. HSD solves this by introducing a hierarchical approach to acceptance probability calculation that better approximates the ideal case while maintaining the lossless property (preserving the target distribution exactly). The paper provides rigorous theoretical analysis proving HSD's correctness and demonstrates consistent improvements over baselines across multiple benchmarks (GSM8K for mathematical reasoning, HumanEval for code generation, and CNN/DailyMail for summarization) using the Qwen2.5 model suite with various scale combinations (0.5B draft model with 14B, 32B, and 72B target models).

**Strengths:**

1. Exceptionally rigorous theoretical analysis with detailed proofs establishing the lossless property of HSD. The paper clearly demonstrates how HSD correctly recovers the target distribution through careful handling of branch divergence and capped ratios.
2. Significant conceptual contribution by identifying and solving the "joint intractability" problem in speculative decoding - how existing methods miscalculate acceptance probabilities for multi-token sequences, leading to suboptimal performance.
3. Elegant hierarchical approach to acceptance probability calculation that provides a theoretically sound solution while remaining practically implementable. The capped ratio concept and unique capping indices are particularly insightful innovations.

**Weaknesses:**

1. Limited experimental scope - the paper only evaluates HSD using Qwen2.5 models across three benchmarks. A broader evaluation with multiple model families and additional tasks would strengthen the empirical validation.
2. Insufficient comparisons with state-of-the-art speculative decoding frameworks. While thoroughly comparing against tokenwise and blockwise methods, the paper omits systematic comparisons with more advanced approachesr.
3. Limited ablation studies to understand the contribution of different HSD components. For instance, how sensitive is performance to the identification of "unique capping indices" or the specific hierarchical structure?

**Questions:**

1. How does the actual computational overhead of HSD (for hierarchical acceptance probability calculations) compare to simpler methods, particularly for smaller draft lengths? Could this overhead offset some gains in block efficiency?
2. The paper mentions HSD can be integrated with multi-draft frameworks but provides limited evaluation. Could you share more comprehensive results comparing HSD Multi-draft against Tokenwise Multi-draft across more model combinations?
4. Have you observed any tasks or model configurations where HSD performs worse than existing methods? Understanding the limitations would help practitioners decide when to use this approach.
5. Could you provide more intuitive examples showing how HSD's hierarchical acceptance mechanism works in practice for concrete generation examples, perhaps with visualizations of the capped ratios and unique capping indices?

---

> ### Author Response · Authors · 2025-11-22
> **Reply to Reviewer ASe8**
>
> We sincerely thank the reviewer for acknowledging our contributions and the invaluable feeback. We address the mentioned questions in detail, as outlined below:
>
> > **W1**: Limited experimental scope - the paper only evaluates HSD using Qwen2.5 models across three benchmarks. A broader evaluation with multiple model families and additional tasks would strengthen the empirical validation.
>
> We expanded our evaluation beyond Qwen2.5 and conducted additional experiments using Llama-3.1-70B-Instruct and Llama-3.1-8B-Instruct pair (non-quantized version, with model weights distributed on 8 H20 GPUs). Evaluation were conducted on four diverse tasks—GSM8K, CNN/DailyMail, HumanEval, and FLORES—covering mathematical reasoning, summarization, code generation, and multilingual translation.
> HSD consistently improves both block efficiency and end-to-end decoding speed over tokenwise and blockwise verification across all tasks, demonstrating that our method generalizes across architectures and domains. Full results are below:
>  #### GSM8K
> | Component         | Block Efficiency | Decoding Speed |
> |-------------------|---------------:|---------------:|
> | Tokenwise         |       6.83         |       8.41|
> | Blockwise         |        7.32       |    8.87|
> | HSD               |         **7.43**        |     **9.18**|
>
> #### CNNdaily
> | Component         | Block Efficiency | Decoding Speed |
> |-------------------|---------------:|---------------:|
> | Tokenwise         |       5.13         |       6.89         |
> | Blockwise         |    5.49            |    7.26            |
> | HSD               |         **5.57**       |      **7.44**          |
>
> #### HumanEval
> | Component         | Block Efficiency | Decoding Speed |
> |-------------------|---------------:|---------------:|
> | Tokenwise         |         8.01       |     13.44           |
> | Blockwise         |        8.33        |  13.56              |
> | HSD               |      **8.40**         |      **13.94**          |
>
> ####  Flores
> | Component | Block Efficiency | Decoding Speed |
> |-----------|-----------------:|---------------:|
> | Tokenwise |             4.92 |           7.81 |
> | Blockwise |             5.05 |           8.04 |
> | HSD       |         **5.22** |       **8.39** |
>
> > **W2**: Insufficient comparisons with state-of-the-art speculative decoding frameworks. While thoroughly comparing against tokenwise and blockwise methods, the paper omits systematic comparisons with more advanced approaches.
>
> We thank the reviewer for the constructive suggestion.
> As clarified in our *Related Work* section, state-of-the-art methods such as **EAGLE-3** rely on a combination of **multi-draft proposal mechanisms** and **self-drafting with fine-tuned layers**, mainly targeting the *drafting* process, whereas our method improves the **verification stage**.  Therefore, they are **not directly comparable**. In fact, HSD integrates smoothly with modern speculative decoding frameworks, since it remains **orthogonal** to techniques that focus on improving the *drafting stage*.
>
> To further demonstrate compatibility with advanced decoding systems, we integrated HSD into the
> state-of-the-art EAGLE-3-LLaMa3.1-Instruct-8B (draft length $\gamma=7$ by default) by replacing its tokenwise verifier:
>
> | Dataset | Method   | Speed (tokens/s) | Block Efficiency (BE) |
> |:--------|:----------|-----------------:|----------------------:|
> | GSM8K   | Baseline |            71.59 |                  3.40 |
> | GSM8K   | +HSD     |        **80.49** |              **3.55** |
>
> These results show that replacing the baseline verifier with HSD yields consistent improvements in both decoding speed and block efficiency. To support reproducibility and benefit the broader community, we have updated our anonymous code repository to include a complete and ready-to-use implementation of HSD integrated with EAGLE-3.
>
> > **W3**: Limited ablation studies to understand the contribution of different HSD components. For instance, how sensitive is performance to the identification of "unique capping indices" or the specific hierarchical structure?
>
> We conducted ablations on the role of the **capped prefix ratio** in Algorithm 2 (HSD),
> which is essential to preserve distributional fidelity.
> Removing capping (i.e., directly using uncapped ratio to compute divergences) yields a slight increase in efficiency,
> but at the cost of **different degrees of performance degradation**（may or may not be obvious depending on the task).
>
> **HSD without Capping**
> | Benchmark | Metric (ACC/BLEU) |         BE |         DS |
> |-----------|------------------:|-----------:|-----------:|
> | GSM8K     |       84.40±1.75% |  6.76±0.05 | 33.63±0.53 |
> | HumanEval |       80.61±0.69% |  5.60±0.06 | 29.15±0.43 |
>
> **HSD**
> | Benchmark | Metric (ACC/BLEU) |        BE |         DS |
> |-----------|------------------:|----------:|-----------:|
> | GSM8K     |       84.96±0.93% | 6.63±0.06 | 32.73±0.55 |
> | HumanEval |       82.47±1.15% | 5.45±0.08 | 27.54±0.48 |

---

> ### Author Response · Authors · 2025-11-22
> **Reply to Reviewer ASe8 — continued**
>
> > **Q1**: How does the actual computational overhead of HSD (for hierarchical acceptance probability calculations) compare to simpler methods, particularly for smaller draft lengths? Could this overhead offset some gains in block efficiency?
>
> We thank the reviewer for raising this question. While there are minor differences in the computational cost of verification—whether using any of the three verification methods—these differences are **insignificant in practice** compared to the reduction in target model forward passes. Indeed, **block efficiency (or equivalently, the acceptance rate) remains the most meaningful metric for evaluating performance**. A detailed complexity analysis is provided in the revised version below:
>
> Both HSD (Eqs. 17–20 in our paper) and blockwise verification (Eqs. 4–5 in [1]) require summing over the vocabulary to compute the acceptance probability at each position. Therefore, HSD introduces **no theoretical overhead** compared to blockwise verification.
>
> Moreover, our implementation (Appendix F) is **more efficient** than that in [1] (Appendix A). By leveraging an equivalent masking mechanism, we eliminate the for-loop and compute probabilities for all positions in parallel. This makes HSD nearly as efficient as tokenwise verification. While tokenwise verification only sums over the vocabulary at a rejected position, **our HSD computation is fully parallelized via tensor operations across both the vocabulary and the draft length $\gamma$, which is typically much smaller than the vocabulary size $\mathcal{V}$**.
>
> Importantly, the computational cost of verification is negligible relative to the reduction in target model forward passes, which is the main bottleneck in verification. Below, we compare the verification cost with the forward-pass reduction for a batch size of 1.
>
> Since the vocabulary size $\mathcal{V}$ is much larger than the draft length $\gamma$, the main cost of HSD arises from computing branch divergences, which requires only $4 \gamma \mathcal{V}$ FLOPs:
>
> 1. $r^*(\mathbf{X}_{1:t}) - 1$ → $\gamma \mathcal{V}$ FLOPs
> 2. Selecting $(r^*(\mathbf{X}_{1:t}) - 1 > 0)$ via the $\max$ operator → $\gamma \mathcal{V}$ FLOPs
> 3. Multiplication by $q$ → $\gamma \mathcal{V}$ FLOPs
> 4. Summation over the vocabulary → $\gamma \mathcal{V}$ FLOPs
>
> For Qwen-2.5 with $|\mathcal{V}| = 151{,}643$ and $\gamma = 10$, this amounts to approximately **5.8M FLOPs**.
>
> In contrast, **the forward-pass FLOPs per new token** in large language models are orders of magnitude larger, even with KV-cache inference. Considering the main contributions:
>
> * $Q \cdot K$ dot products
> * Attention score $\times V$
> * All projection/MLP FLOPs
> * Ignoring softmax, LayerNorm, rotary embeddings, and bias
>
> The per-token FLOPs can be approximated as:
>
> $$
> \text{FLOPs per new token} \approx L \cdot \Big[4 d L_{\text{past}} + 4 d^2 + 2 d d_{\text{ff}}\Big]
> $$
>
> Where:
>
> * $L$ = number of transformer layers
> * $d$ = hidden size
> * $d_\text{ff}$ = intermediate (MLP) size
> * $L_{\text{past}}$ = number of cached past tokens
>
> Using a context length $L_{\text{past}} = 1024$, the per-token FLOPs for Qwen2.5 models are:
>
> * Qwen2.5-0.5B: **0.374 GFLOPs**
> * Qwen2.5-72B: **62.915 GFLOPs**
>
> As shown in Table 1 in our paper, all methods achieve block efficiency larger than 2. Consequently, the cost of HSD verification is negligible relative to the reduction in target model forward passes.
>
> To directly quantify the overhead of the verification stage, we evaluate verification cost on 100 GSM8K problems using GPTQ-quantized 8-bit Qwen2.5-72B-Instruct and Qwen2.5-0.5B-Instruct as target and draft models on a single H200 GPU. Across all settings we consider, the verification stage consistently accounts for **less than 1%** of the total decoding time. At the same time, the vast majority of runtime is spent in the draft and target forward passes. Here, the draft forward pass accounts for about 24% of the runtime, and the target forward pass accounts for about 72% in both blockwise and HSD. The verification stage of HSD is about 20% faster than that of blockwise.
>
> **Qwen2.5-72B Overall Components**
>
> | Component         | Blockwise Mean (ms/token) | Blockwise % of Total | HSD Mean (ms/token) | HSD % of Total |
> |------------------|---------------------------:|----------------------:|---------------------:|----------------:|
> | **Total**            | **34.168** | **100.00%** |**33.788**|**100.00%**|
> | Prefill          | 0.913 | 2.67% | 0.915 | 2.71% |
> | Draft Forward    | 8.210 | 24.03% | 7.865 | 23.28% |
> | Target Forward   | 24.690 | 72.26% | 24.695 | 73.09%|
> | KV Cache Input   | 0.007 | 0.02% | 0.005 | 0.01%|
> | KV Cache Output  | 0.070 | 0.20% | 0.067 | 0.20%|
> | Logits Processing| 0.093 | 0.27% | 0.103 | 0.31%|
> | Verification     | 0.160 | 0.47% | 0.127 | 0.37%|
> | Other            | 0.025 | 0.08% | 0.012 | 0.03%|

---

> ### Author Response · Authors · 2025-11-22
> **Reply to Reviewer ASe8 — continued**
>
> > **Q2**: The paper mentions HSD can be integrated with multi-draft frameworks but provides limited evaluation. Could you share more comprehensive results comparing HSD Multi-draft against Tokenwise Multi-draft across more model combinations?
>
> To address **Q2**, we provide an expanded set of **multi-draft evaluations** comparing **HSD Multi-draft** to **Tokenwise Multi-draft**, using a different model family from the Qwen experiments in the main paper. Specifically, we evaluate the **Llama-3.1-70B-Instruct / Llama-3.1-8B-Instruct** pair (non-quantized), with model weights distributed across 8 H20 GPUs. This follows the multi-draft setup described in Section 6 of the paper.
>
> **GSM8K**
>
> | Benchmark |       BE |        DS |
> |-----------|---------:|----------:|
> |    Tokenwise  |     8.72 |     10.21 |
> |    HSD  | **9.00** | **11.02** |
>
> To further support community adoption of HSD in multi-draft settings, we also provided an extended implementation of the HuggingFace transformers framework that natively supports multi-draft decoding, which is available in our anonymous codebase.
>
> > **Q3**: Have you observed any tasks or model configurations where HSD performs worse than existing methods? Understanding the limitations would help practitioners decide when to use this approach.
>
> As guaranteed by **Theorem 5**, HSD is at least as efficient as token-wise verification in all cases. Its benefits are most noticeable in settings with high uncertainty—such as when the sampling distribution has large entropy or when there is substantial divergence between the target and the draft distribution.
>
> - When **greedy decoding is used for the target model**, which collapses the distribution to its argmax, all lossless verification methods reduce to the same pattern-matching behavior.
> - As the target distribution becomes closer to the draft distribution, differences among lossless verification methods diminish, making HSD’s gains less pronounced.
>
> > **Q4**: Could you provide more intuitive examples showing how HSD's hierarchical acceptance mechanism works in practice for concrete generation examples, perhaps with visualizations of the capped ratios and unique capping indices?
>
> Thanks for your insightful advice! To demonstrate how HSD operates, we consider the following GSM8K question:
> ```
> Eliza's rate per hour for the first 40 hours she works each week is $10. She also receives an overtime pay of 1.2 times her regular hourly rate. If Eliza worked for 45 hours this week, how much are her earnings for this week?
> ```
> and the following generated prefix:
> ```
> To determine Eliza's earnings for the week, we need to calculate both her regular pay and her overtime pay.
>   1. **Calculate Regular Pay:**
>   - Eliza's regular rate is $10 per hour.
>   - She worked 40 hours at her regular rate.
>   - Regular pay = 40 hours * $10/hour = $400.
> - 2. **Calculate Overtime Pay:**
>   - Eliza worked a total of 45 hours, so
> ```
> Under this prefix, the small and large models respectively produce the probability vectors ($\gamma=10$):
>   - $\{p(\cdot),..., p(\cdot|\mathbf{X}_{1:\gamma})\}$ = $\{0.7156, 1.0000, 1.0000, 1.0000, 0.0000, 0.0000, 0.0000, 1.0000, 0.0000, 0.4968\}$
>   - $\{q(\cdot),..., q(\cdot|\mathbf{X}_{1:\gamma})\}$= $\{0.8771, 0.7900, 0.6514, 0.2592, 0.6773, 0.1490, 0.5775, 1.0000, 0.4611, 0.3630\}$
>
> The corresponding draft is: $\{she, work, ed, 45, -, 40, =, 5, hours, of\}$
>
> First, we compute the joint probabilities:
>   - $\{p\left(\mathbf{X}_{1:t}\right) \, \text{for} \, t=1, \dots, \gamma\} = \{0.7156, 0.7156, 0.7156, 0.7156, 0, 0, 0, 0, 0, 0\}$
>   - $\{q\left(\mathbf{X}_{1:t}\right) \, \text{for} \, t=1, \dots, \gamma\} = \{0.8771, 0.6929, 0.4513, 0.1170, 0.0792, 0.0118, 0.0068, 0.0068, 0.0031, 0.0011\}$
>
> The ratio of joint probabilities is: $\{r(\mathbf{X}_{1:t})\, \text{for} \, t=1, \dots, \gamma\}  = \{0.8159, 1.0327, 1.5855, 6.1171, 0, 0, 0, 0, 0, 0\}$
>
> According to **Definition 4**, we obtain the maximum prefix indices:
>   - $\{m(\mathbf{X}_{1:t})\, \text{for} \, t=1, \dots, \gamma\} = \{0, 0, 2, 3, 4, 4, 4, 4, 4, 4\}$
>
> Based on the full vocabulary distributions (see **Eq.(16-18)**), we compute:
>   - $\{D^*_{\text{Branch}}\left(p, q \mid \mathbf{X}_{1: t}\right)\, \text{for} \, t=1, \dots, \gamma\}  = \{0.0227, 0.2416, 0.3343, 0.0991, 0, 0, 0, 0, 0, 0\}$
>   - $\{D^*_{\text{Branch}}\left(q, p \mid \mathbf{X}_{1: t}\right)\, \text{for} \, t=1, \dots, \gamma\} = \{0.1842, 0.2416, 0.3343, 0.0991, 0.0792, 0.0118, 0.0068, 0.0068, 0.0031, 0.0011\}$
>
> According to **Eq.(19)**, we obtain HSD’s acceptance probabilities: $h_t$ (HSD) = $\{0.1231, 1, 1, 1, 0, 0, 0, 0, 0, 0\}$
>
> Since HSD judges from back to front, it will accept all the first 4 tokens, yielding: $n_{\text{match}} = 4$
>
> If we adopt the tokenwise verification baseline, we obtain: $h_t$ (tokenwise) = $\{0.8159, 1, 1, 1, 0, 0, 0, 1, 0, 1\}$
>
> Since this method checks sequentially from front to back, with $h_1 = 0.8159$ there is a chance of rejection, causing the entire draft to be discarded.

---

### Author Response · Authors · 2025-12-03
**Summary of Revisions**

We sincerely thank all reviewers for their constructive and insightful feedback. We have carefully incorporated their suggestions into our revised manuscript. Specifically, we have:

- **Extended our experimental evaluation** by including results on additional benchmarks and models, integration/comparison with state-of-the-art method, and an ablation study on the key capping mechanism, as suggested by **Reviewer ASe8** and **Reviewer JMzF**. These experiments are now included in **Appendix E**.

- **Expanded our computational efficiency analysis** by providing a detailed complexity derivation and a comprehensive runtime breakdown comparing HSD with other verification methods, following the recommendations of **Reviewer ASe8**, **Reviewer YeUo**, and **Reviewer JMzF**. These additions appear in **Appendix H**.

- **Added a concrete generation example** with a step-by-step, intuitive explanation, in response to comments from **Reviewer ASe8**, **Reviewer YeUo**, and **Reviewer qE7w**. This example provides an intuitive complement to the formal mathematical exposition of the method and is now included in **Appendix I**.

We hope these updates address the reviewers’ concerns and further improve the clarity and completeness of our work.

---

### Meta-Review · Area_Chair_jXV3 · 2025-12-20

**Summary:**

**Paper summary.** This paper studies speculative decoding verification and argues that existing tokenwise/blockwise verification struggles to accept long drafts because of a “joint intractability” issue. The authors propose Hierarchical Speculative Decoding (HSD), which verifies drafts using a hierarchical resampling procedure while keeping the sampling distribution *lossless* (i.e., matching the target model). The paper includes formal proofs and shows improved block efficiency / throughput over tokenwise and blockwise verification in the reported experiments.

**What happened in the discussion.** Reviewers mostly agreed the theory is the core value here. The main pushback was (1) the algorithm is hard to understand quickly from the math, (2) the original experiments were mostly limited to one model family and a small set of tasks, and (3) HSD may add extra per-step computation (ratios / backward scan), so readers want clearer wall-time breakdowns. In the forum, the authors responded with (a) additional experiments on another model family (including a multilingual task), (b) more detail on computational overhead and why it is small compared to saved target forward passes, and (c) additional multi-draft results and clarifications for the algorithm flow.

**My assessment as AC.** After reading the paper and the full forum record, I think the contribution is real: the paper makes a clear claim (“lossless verification that accepts longer drafts”) and backs it up with proofs and consistent speed/efficiency gains. The remaining weaknesses are mainly presentation and completeness of comparisons, not correctness. I agree with reviewers that the final paper should add a simple pseudocode/diagram and a short wall-time breakdown table, but I do not see a blocking technical flaw.

**Decision.** Accept (Oral). The paper has a solid theoretical contribution and the authors addressed the biggest scope/overhead concerns with additional evidence during the discussion.

**Reviewer Concerns:**

- **Reviewer YeUo (rating 8, confidence 4)**: Asked for a clearer intuitive explanation of why the hierarchical procedure helps and a small algorithm detail check; authors added an intuitive example and clarified algorithm details. **Status:** mostly resolved.
- **Reviewer ASe8 (rating 6, confidence 5)**: Raised (i) limited experimental scope, (ii) missing comparisons to more advanced speculative decoding frameworks, and (iii) limited ablations. Authors added experiments on another model family and added more multi-draft results; comparisons are still not exhaustive. **Status:** scope concern resolved; comparisons/ablations partially resolved.
- **Reviewer JMzF (rating 6, confidence 3)**: Asked for broader evaluation and a clearer complexity/latency breakdown. Authors added broader evaluation and provided a more explicit overhead discussion. **Status:** largely resolved, with minor remaining request for a compact wall-time table.
- **Reviewer qE7w (confidence 3; rating not filled in review form)**: Main concern is that the algorithm steps are buried in dense math and the flow is hard to follow. Authors added more explanation and examples. **Status:** partially resolved; still needs a “one-glance” description in the final paper.

**Reviewer Scores:**

- **YeUo (8,4)**: Likely unchanged.
- **ASe8 (6,5)**: Likely unchanged or slightly higher after added experiments; remaining concerns are not blocking.
- **JMzF (6,3)**: Likely unchanged.
- **qE7w (conf 3)**: I expect an improvement if the paper adds clearer pseudocode/diagram, but cannot infer a specific score since the rating was not provided.

---

### Decision · Program_Chairs · 2026-01-26

Accept (Oral)